# AMPK activation by glycogen expenditure primes the exit of naïve pluripotency

Seong-Min Kim[1,5], Eun-Ji Kwon[1,5], Ji-Young Oh[1], Han Sun Kim [ID][2], Sunghyouk Park [ID][2], Goo Jang[3], Jeong Tae Do [ID][4], Keun-Tae Kim [ID][1✉] & Hyuk-Jin Cha [ID][1✉]

## Abstract

**Embryonic and epiblast stem cells in pre-and post-implantation embryos are characterized by their naïve and primed states, respectively which represent distinct phases of pluripotency. Thus, cellular transition from naïve-to-primed pluripotency recapitulates a drastic metabolic and cellular remodeling after implantation to adapt to changes in extracellular conditions. Here, we found that inhibition of AMPK occurs during naïve transition with two conventional inhibitors of the MEK1 and GSK3β pathways. The accumulation of glycogen due to iGSK3β is responsible for AMPK inhibition, which accounts for high de novo fatty acid synthesis in naïve (ESCs). The knockout of glycogen synthase 1 in naïve ESCs; GKO, resulting in a drastic glycogen loss, leads to a robust AMPK activation and lowers the level of fatty acids. GKO loses cellular characteristics of naïve ESCs and rapidly transitioned to a primed state. The characteristics of GKO are restored by the simultaneous AMPK KO. These findings suggest that high glycogen in epiblast within pre-implantation blastocyst may act as a signaling molecule for timely activation of AMPK, thus ultimately contributing to transition to post-implantation stage epiblast.**

**Keywords** naïve Pluripotency; Glycogen; AMPK; Fatty Acids; Pre-implantation Embryo
**Subject Categories** Metabolism; Stem Cells & Regenerative Medicine

## Introduction

The unique cellular characteristics of naïve and primed embryonic stem cells (ESCs) such as signaling, epigenetics, and metabolism reflect the biology of embryonic and epiblast stem cells of pre- and post-implantation embryos (Kim et al, 2022c; Nichols and Smith, 2009; Shan et al, 2017). The requirement of two chemical inhibitors (2i) for MEK1/2 and GSK3β pathways for maintaining naïve ESCs epitomizes the constant suppression of MEK1 and GSK3β pathways

by Netrin-1 signaling in the preimplantation embryo (Huyghe et al, 2020). Additionally, the high dependency of fatty acids on the survival of ESCs (Tanosaki et al, 2020; Yan et al, 2021) illustrates the critical roles of fatty acids during early embryogenesis, as demonstrated by the embryo lethality caused by genetic perturbation of key genes involved in fatty acid synthesis such as fatty acid synthase (*Fasn*) (Chirala et al, 2003) or acetyl-CoA carboxylase 1 (*Acc1*) (Abu-Elheiga et al, 2005). Thus, the active production of fatty acid up to the blastocyst stage (Haggarty et al, 2006) suggests that embryos favor fatty acid oxidation (FAO) as a source of energy (Krisher and Prather, 2012).

Particularly, the dynamic change of glucose metabolism during embryo development has also been closely characterized by comparing naïve and primed ESCs [reviewed in (Folmes et al, 2012; Perestrelo et al, 2018)]. Although primed ESCs and human induced pluripotent stem cells (iPSCs) predominantly depend on glycolysis (Cha et al, 2017), a clear metabolic shift to bivalent metabolism with capacity for oxidative phosphorylation (OxPHOS) occurs in naïve ESCs (Sperber et al, 2015; Zhou et al, 2012). Such metabolic transition (i.e., primed to naïve) is induced by epigenetic changes from differentially produced metabolites such as attenuation of H3K27me3 by decrease of *S*-adenosyl methionine (Sperber et al, 2015) and increase of α-ketoglutarate (Carey et al, 2015). Interestingly, the metabolic transition from primed to naïve ESCs is initiated by simultaneous inhibition of MEK1 and GSK3β with 2i (Kim et al, 2022a; Koo et al, 2023; Vardhana et al, 2019) through the upregulation of *Esrrb* (Sone et al, 2017) or *LIN28* (Zhang et al, 2016). *Esrrb* expression is a downstream event of GSK3β inhibition (Martello et al, 2012), and therefore signaling pathways altered by 2i determine the acquisition of naïve pluripotency.

Glycogen dynamics during embryonic development have been characterized for decades (Edirisinghe et al, 1984a; Thomson and Brinster, 1966) and its role as a stored energy source has been previously reported (Dean, 2019). Particularly, glycogen levels increase with embryonic development from the 2-cell stage to the early blastocyst stage, which is followed by a sharp decrease after the late blastocyst stage (Edirisinghe et al, 1984b; Thomson and Brinster, 1966). Although GSK3β phosphorylates and inhibits glycogen synthase (encoded by *Gys1*), exposure to a GSK3β inhibitor to consequently activate glycogen synthase only promotes

[1]College of Pharmacy and Research Institute of Pharmaceutical Sciences, Seoul National University, Seoul, Republic of Korea. [2]Natural Products Research Institute, College of Pharmacy, Seoul National University, Seoul, Republic of Korea. [3]Laboratory of Theriogenology and Biotechnology, Department of Veterinary Clinical Sciences, College of Veterinary Medicine, Seoul National University, Seoul, Republic of Korea. [4]Department of Stem Cell and Regenerative Biotechnology, KU Institute of Science and Technology, Konkuk University, Seoul, Republic of Korea. [5]These authors contributed equally: Seong-Min Kim, Eun-Ji Kwon. ✉E-mail: keuntaekim91@gmail.com; hjcha93@snu.ac.kr

glycogen synthesis in naïve ESCs (Kim et al, 2022a), which is reproduced in human ESCs (Chen et al, 2015). These studies suggest that naïve specific glycogen itself has roles in maintaining naïve pluripotency (Chen et al, 2015; Kim et al, 2022a).

This study aimed to determine the role of intracellular glycogen, which we had previously found to exclusively exist in naïve pluripotency. Through the establishment of a *Gys1* knockout, we demonstrated that intracellular glycogen would serve as a signaling molecule for the timely activation of AMPK to repress the de novo synthesis of fatty acids, whose temporal reduction primes an exit from naïve pluripotency.

# Results

## Reduced AMPK-dependent phosphoproteome during the transition to the naïve state with 2i

The induction of naïve pluripotency [or ground state as previously demonstrated (Galonska et al, 2015; Marks et al, 2012)] to mimic the pre-implantation embryos is achieved through simultaneous inhibition of MEK1 and GSK3β with chemical inhibitors (hereinafter referred to as 2i) along with leukemia inhibitory factor (LIF). The timely alteration of the transcriptome during this transition provides a snapshot of the changes in gene expression patterns in response to these agents (Yang et al, 2019). However, the kinase signaling pathways that initiate such gene response upon simultaneous inhibition of MEK1 and GSK3β remain largely unexplored. To this end, we took advantage of a recently published phosphoproteome dataset collected at multiple stages of naïve pluripotency induction (Martinez-Val et al, 2021); Data ref: Martinez-Val et al, 2021), as shown in Fig. 1A. Through a meta-analysis of the phosphoproteome datasets, we identified four clusters with distinct phosphoproteome signatures (SL: Cont, 0.5 h, 1 h/2 h, and 6 h after 2i treatment) (Fig. 1B). The differentially phosphorylated proteins (DPs) became manifested from 0.5 h (Fig. 1C) to 6 h after 2i treatment (Fig. EV1A). To investigate the perturbation-induced changes, we employed hierarchical clustering to group phosphoproteins based on their phosphorylation patterns across time points. Subsequently, we conducted an analysis to determine upstream kinases corresponds to phosphoproteins within each cluster (Fig. 1D). As anticipated, phosphoproteins in cluster 3 exhibited a gradual decrease in phosphorylation over time following 2i treatment, showing associations with downstream substrates of MAPK1, ERK1, and GSK3, as a result of iMEK1 and iGSK3β treatment (Fig. EV1B). Concurrently, the observed reduction in phosphorylation of phosphoproteins also correlated with AMPK downstream substrates (Fig. 1E). In addition, we analyzed phosphoproteome data from temporal naïve pluripotency induction, independently achieved by Cdk8/19 inhibitor treatment (Cdk8i) (Fig. EV1C) (Martinez-Val et al, 2021). Similar to the case of 2i treatment for inducing naïve pluripotency, the time-dependent phosphoproteome profiles of Cdk8i treatment revealed that cluster 1 phosphoproteins showed a consistent reduction pattern, which overlapped with downstream substrates of ERK1 and MAPK1 (Fig. EV1D). Moreover, AMPK substrates were also enriched in cluster 1 (Fig. EV1E). These findings suggest a strong association between AMPK pathway inhibition and the induction of a naïve pluripotent state.

## A high level of fatty acid in naïve ESCs is concurrent with AMPK inhibition

Given the significant reduction in the phosphoproteome of AMPK substrates during naïve pluripotency induction (Fig. 1), inhibition of both MEK1 and GSK3β may lead to the repression of AMPK activity. LIF treatment activates both STAT3 and MEK1/2, which subsequently leads to ERK1/2 phosphorylation (Kim et al, 2022b). Meanwhile, GSK3β, an upstream kinase responsible for the inhibitory phosphorylation of glycogen synthase (GYS1), remains constitutively active via autophosphorylation at tyrosine 216 (Beurel et al, 2015), thereby continuously inhibiting GYS1 activity (Fig. 2A). As a result, 2i treatment (i.e., iMEK1 and iGSK3β) reduces the phosphorylation levels of ERK1/2 and GYS1 (Fig. 2A). As expected, a marked reduction in the phosphorylation of ERK1/2, GSK3β, and GYS1 was observed following 2i treatment. In contrast, primed ESCs exhibited distinct active phosphorylation of GSK3β [pGSK3β(A)] and ERK1/2, as well as inhibitory phosphorylation of GYS1 [pGYS1(I)], corresponding to the phosphorylation-dependent electrophoretic mobility shift (PDEMS) of GYS1 (blue arrows in Fig. 2B). In the distinct signaling environments of naïve and primed ESCs, AMPK activation in primed ESCs was indicated by the active phospho-AMPK signal [pAMPKα(A)] and the emerging inhibitory phosphorylation of acetyl-CoA carboxylase (ACC) [pACC(I)], a typical downstream target of AMPK (Steinberg and Carling, 2019) (Fig. 2B–D). Notably, ACC activity, tightly regulated by AMPK in response to cellular energy levels, plays a crucial role in determining intracellular fatty acid levels by promoting de novo fatty acid synthesis while repressing fatty acid β-oxidation (FAO). Thereby, ACC inhibition accompanied with AMPK activation (Fig. 2B–D) likely contributes to the significant reduction of total fatty acid observed in primed ESCs compared to naïve ESCs, as determined by Nile Red staining (Fig. 2E). To identify the underlying mechanisms of the reduction of fatty acids in primed ESCs, we next examined the de novo fatty acid synthesis using $^{13}$C glucose (Fig. 2F). The newly synthesized fatty acids from $^{13}$C glucose, that were determined by the split peaks of the omega-methyl signal on the NMR spectrum (Wen et al, 2019), were much higher in naïve ESCs (Fig. 2G). Deprivation of glucose significantly reduced the intensity of BODIPY staining in naïve ESCs, suggesting that active de novo fatty acid synthesis is responsible for high fatty acid levels (Fig. 2H). Unlike primed ESCs, which are committed to differentiation, naïve ESCs are more inclined to maintain pluripotency (Nichols and Smith, 2009; Nichols and Smith, 2012; Ying et al, 2008). Therefore, primed ESCs are typically cultured with serum replacement (SR) to avoid differentiation stimuli, while naïve ESCs, which are more resistant to differentiation signals, are often cultured with fetal bovine serum (FBS). To eliminate any bias from differing culture conditions, a set of naïve and primed ESCs was maintained in the same culture medium—either 2i+LIF for naïve ESCs or bFGF2+Activin A for primed ESCs. Culturing both naïve and primed ESCs in the identical N2B27 medium resulted in only marginal differences in colony morphology (Fig. EV2A), intracellular glycogen levels (Fig. EV2B), and inhibitory phosphorylation of GYS1 [pGYS1(I)] and ACC [pACC(I)] (Fig. EV2C). This suggests that the distinct metabolic differences between naïve and primed ESCs are not solely due to the dissimilarities in their culture media.

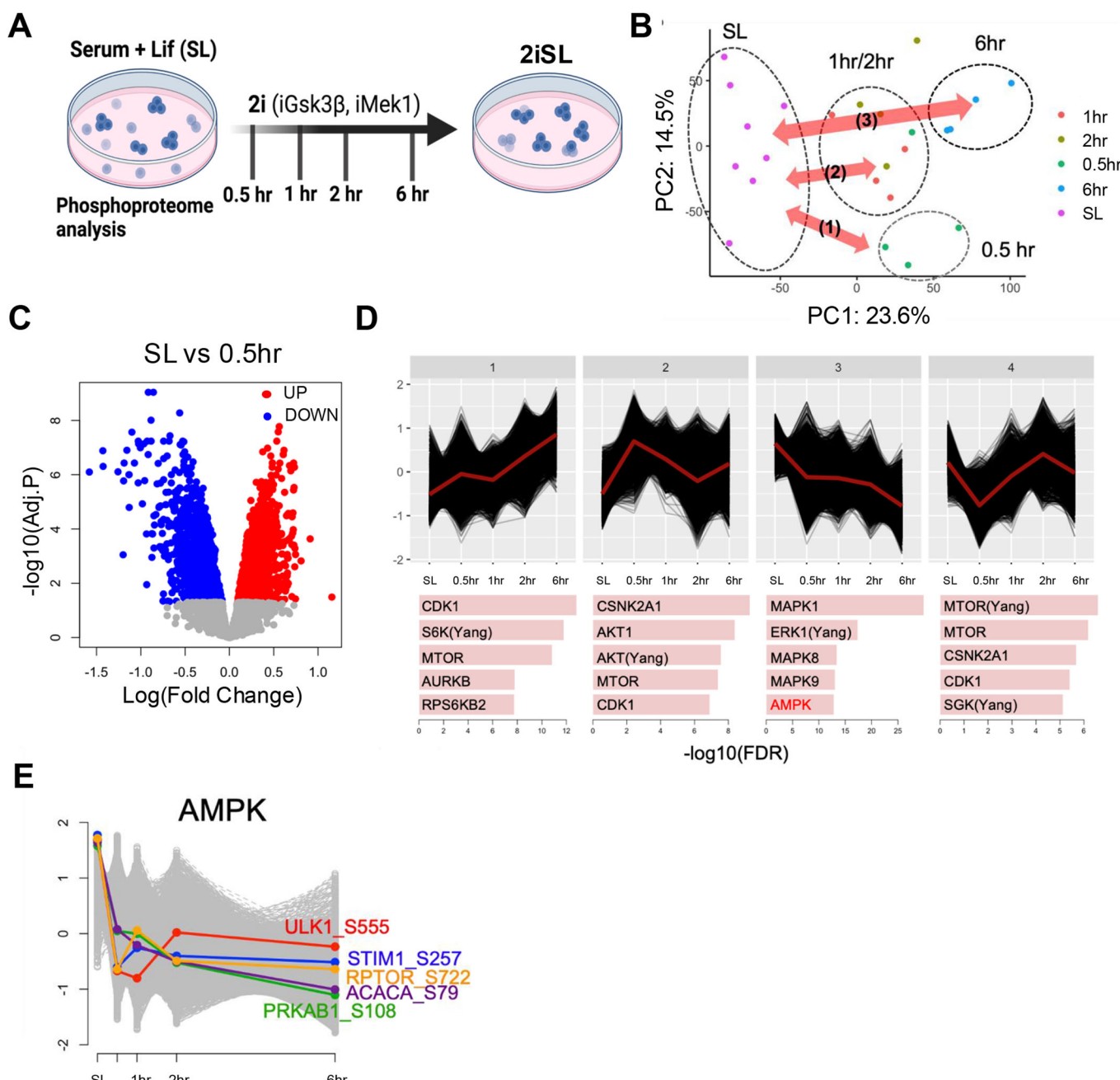

**Figure 1. Reduced AMPK-dependent phosphoproteome during the transition to the naïve state with 2i.**

(A) Graphical image of experimental flow from Mariniez-Val et al, 2021, (B) Principal component analysis of the phosphoproteome from (A). (C) Volcano plot of differentially phosphorylated proteins after 0.5 h of treatment ($n = 4$) compared to SL ($n = 8$). We employed moderated t-test, which was implemented in the R package limma (v3.62.1). Differentially phosphorylated proteins were selected based on a Benjamini–Hochberg-adjusted P-value less than 0.05 and an absolute fold-change greater than 0 as the cut-offs. (D) Hierarchical clustering results representing four phosphoprotein' clusters showing distinct phosphoproteome changes at various time points. The top five kinase categories significantly enriched in each phosphoprotein cluster. (E) Phosphoproteins phosphorylated by AMPK among the phosphoproteins belonging to cluster 3. Source data are available online for this figure.

The higher reactivity of the inner cell mass of E3.5 mouse blastocysts to Nile Red also further supports the exclusive occurrence of fatty acids in naïve ESCs (Fig. 2I). Consistently, the inhibition of fatty acid synthesis with a chemical inhibitor of Acc, induced drastic cell death in naïve ESCs (Fig. 2J). Overall, these results indicate that the constant inactivation of AMPK and the concurrent elevation in fatty acid level are typical features of naïve mESCs.

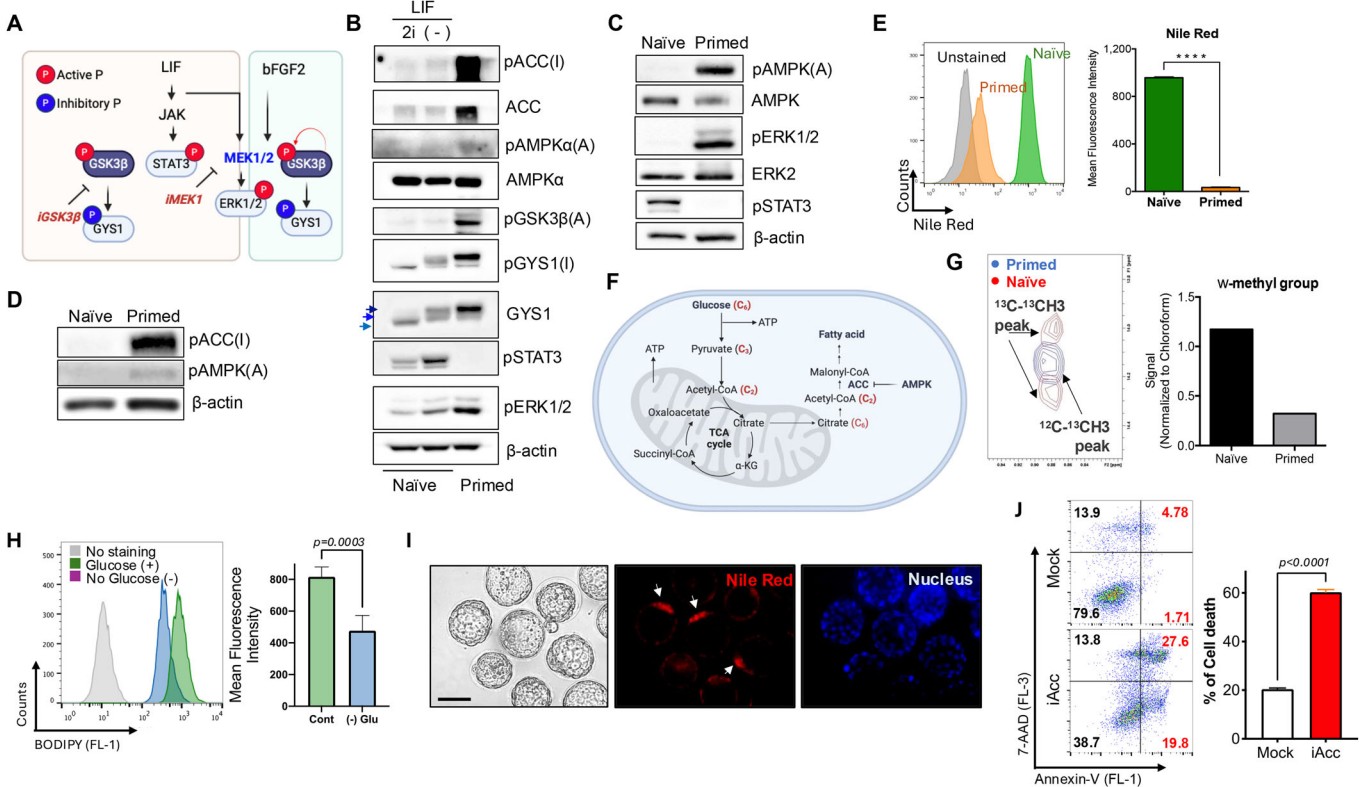

**Figure 2. A high level of fatty acid in naïve ESCs is concurrent with AMPK inhibition.**

(A) Graphical summary of naïve conversion (LIF/iGSK3β/iMEK1 [2i]) and primed conversion (Fgf2/Activin A [FA]) from intermediate status (LIF/Serum). (B) Immunoblotting analysis for indicative proteins (pAMPKα, AMPK, pErk1/2, pGSK3β, pGYS1, pSTAT3) in J1 (naïve) and PJ1 (primed) mESCs, β-actin was used as a loading control. A and I from phosphorylated proteins indicate active [A] or inactive [I] phosphorylation, respectively. Blue arrows indicate phosphorylation-dependent electrophoretic mobility shift (PDEMS) of GYS1. (C) Immunoblotting analysis for indicative proteins (pERK1/2, pAMPKα, pSTAT3 in J1 (naïve) and PJ1 (primed) mESCs), β-actin was used as a loading control. (D) Immunoblot analysis of J1 (naïve) and PJ1 (primed) mESCs with indicative proteins (pACC, pAMPK), β-actin was used as a loading control. (E) Flow cytometry of Nile Red staining of J1 (naïve) and PJ1 (primed) mESCs (left panel), quantification of mean fluorescence intensity from the flow cytometry (right panel). Multiple t-tests, (****$P < 0.0001$, $n = 3$). This bar graph shows mean value with standard deviation as an error bar. (F) Graphical summary of overall glucose metabolism containing glycolysis, TCA cycle and de novo fatty acid synthesis. (G) (Left) $^1$H-$^{13}$C Heteronuclear Single Quantum Coherence (HSQC) NMR spectra for Primed (Blue) and Naive (Red) cells in the region of ω-methyl group of fatty acids. The split peaks indicate the last two carbons of the fatty acid tail are all $^{13}$C-labeled. (Right) The signal intensity of the split peaks in Primed and Naive cells in the left, normalized by the signal intensity of the solvent chloroform of each sample. (H) Flow cytometry of BODIPY 493/503 staining of J1 with [Cont] or without [(−)Glu] glucose. Glucose was depleted for 24 h before the analysis and quantification of mean fluorescence intensity from the flow cytometry (right panel). Multiple t-tests, ($N = 3$, $n = 3$). This bar graph shows mean value with standard deviation as an error bar. (I) Mouse embryos (3.5-4.5dpc) were collected fixed with 4% PFA followed by 1 μg/mL of Nile Red suspended for 1 h at 37°C. Hoechst was used for nucleus counterstaining of embryos. White arrows indicate inner cell mass part in each embryo. Scale bar, 100 μm ($N = 2$, $n = 3$). See also Fig. 3C. (J) Cell viability after iACC treatment on mESCs was measured through flow cytometric analysis of Annexin-V/7-AAD staining Multiple t-tests, ($N = 3$, $n = 3$). Source data are available online for this figure.

## High intracellular glycogen represses AMPK activation in naïve ESCs

The loss of typical dome-shape colony morphology upon the deprivation of either iMEK1 or iGSK3β (Kim et al, 2022c), highlights the requirement of both iMEK1 and iGSK3β in maintaining naïve pluripotency (Marks et al, 2012). AMPK activation was observed upon the withdrawal of iGSK3β in naïve ESCs (Fig. 3B), suggesting that inhibition of GSK3β dependent signaling represses AMPK activation. Of note, the inhibition of AMPK by iGSK3β treatment is interesting because GSK3β typically inhibits AMPK activity by directly phosphorylating its regulatory domain (Suzuki et al, 2013). This observation suggests that there may be other unknown factors involved in the observed AMPK

inhibition in naïve ESCs when treated with iGSK3β. On the other hand, previous studies have reported that intracellular glycogen can inhibit AMPK activity by directly interacting with the glycogen binding domain (CBD) of the AMPKβ subunit (McBride et al, 2009). Notably, inhibition of GSK3β has been demonstrated to stimulate glycogen production in naïve ESCs, while the depletion of GSK3β inhibitor significantly diminishes glycogen stores in naïve ESCs but does not affect primed ESCs (Kim et al, 2022a). This suggests that the production of glycogen in naïve ESCs upon GSK3β inhibition may be a contributing factor to the observed inhibition of AMPK activity in these cells. Naïve-specific glycogen production by iGSK3β as described in our previous study (Kim et al, 2022a) was highlighted by clear staining of CDg4, a fluorescent probe for glycogen (Allott et al, 2020; Kim et al,

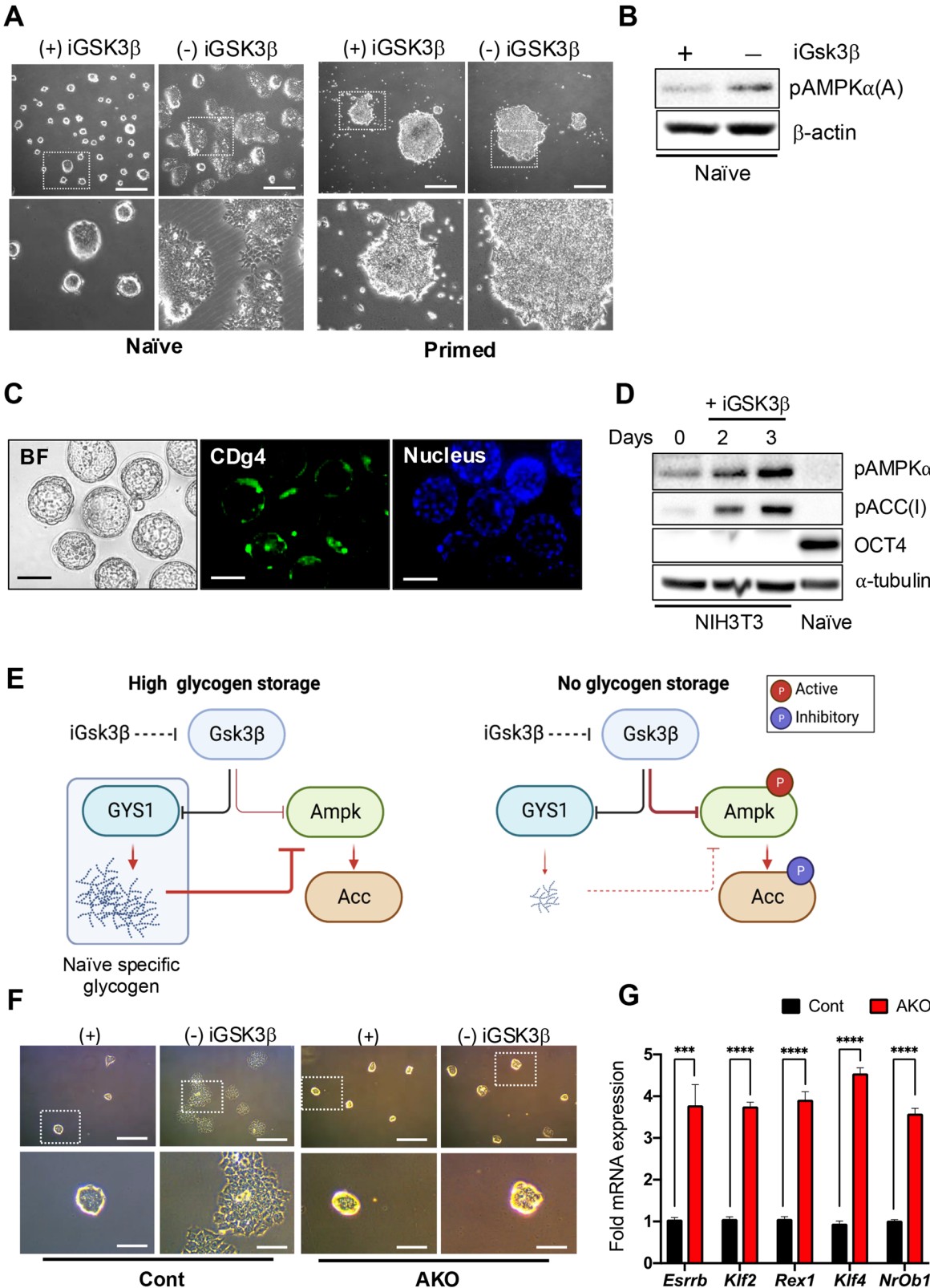

◄

**Figure 3.  High intracellular glycogen represses AMPK activation in naïve ESCs.**

(A) Representative brightfield images of J1 (naïve) and PJ1 (primed) mESCs with or without iGSK3β treatment (scale bar = 500 μm). (B) Immunoblotting analysis for indicative proteins (pAMPKα in J1 (naïve) and PJ1 (primed) mESCs with or without iGSK3β treatment). See also Fig. EV4G. (C) 3.5-4.5dpc of mouse blastocysts were collected and fixed with 4% PFA. 3 mM of CDg4 was treated for 1 h for embryo staining. Hoechst for nucleus counterstaining. Scale bar, 100 μm. See also Fig. 2i (N = 2, n = 3). (D) Immunoblotting analysis for indicative proteins (pAMPKα, pAcc, Oct4 in J1 (naïve) and NIH3T3 under indicative conditions), α-tubulin was used for the loading control. (E) Graphical abstract of intracellular signaling mechanism for AMPK and ACC activity under high glycogen storage condition (left panel) or no glycogen storage condition (right panel). Cont: Control, AKO: *Prkaa1* KO. (F) Representative brightfield images of J1 Cont and J1 AKO mESCs with or without iGSK3β treatment (scale bar = 500 μm). (G) Relative mRNA expression of J1 Cont and J1 AKO mESCs with naïve pluripotency genes (*Esrrb, Klf2, Rex1, Klf4* and *NrOb1*) were quantified with RT-qPCR analysis. Each gene expression was normalized with internal control gene, *Gapdh*. Multiple t-tests, (****$P < 0.0001$, ***$P = 0.0006$, N = 3, n = 3). This bar graph shows mean value with standard deviation as an error bar. Source data are available online for this figure.

2022a; Lee et al, 2012) in the inner cell mass (ICM) of a mouse blastocyst (Fig. 3C), which was consistent with previous observations (Thomson and Brinster, 1966). In sharp contrast to mESCs, the activation of AMPK by GSK3β inhibition (with iGSK3β) occurred in NIH3T3 cells (i.e., normal somatic cells) where glycogen was obviously lacking (Figs. 3D and EV2A). Thus, we concluded that naïve specific glycogen production by GSK3β inhibition could explain the contradictory repression of AMPK activity in naïve ESCs (Fig. 3E). Based on the hypothesis that AMPK activation by removal of iGSK3β contributes to the loss of pluripotency, further experiments were conducted using *Prkaa1* (AMPK) knockout (AKO) naïve ESCs (Fig. EV2B). Interestingly, the AKO retained the typical dome-shaped morphology even after depletion of iGSK3β, whereas the control group largely lost this characteristic dome shape (Fig. 3F). This suggests that AMPK activation may play a role in the loss of naïve pluripotency induced by iGSK3β depletion, and its absence through knockout may prevent the morphological changes associated with loss of naïve pluripotency in response to iGSK3β depletion. Consistently, typical naïve marker genes were highly expressed in AKO compared to the control (Fig. 3G).

## Role of glycogen in the suppression of AMPK activation and maintaining fatty acid levels

We next sought to determine the role of glycogen in naïve ESCs. To this end, we first established a line of naïve ESCs lacking glycogen by introducing an indel (insertion and deletion) at exon 9 of glycogen synthase 1 (*Gys1*) (Fig. 4A). After conducting the T7E1 assay to confirm the knockout of *Gys1* in naïve ESCs (Fig. 4B), multiple single clones of *Gys1* KO naïve ESCs (hereinafter referred to as GKO) were established. As expected, intracellular glycogen was completely depleted in the GKO (Fig. 4C) and CDg4 positive naïve ESCs population was markedly reduced (Fig. 4D), while the cell cycle profile remained comparable (Fig. EV4A). The temporary resistance to glucose deprivation in naïve ESCs, which was accompanied by a loss of stored glycogen (Kim et al, 2022a), was markedly reduced in GKO (Fig. EV4B). In parallel with glycogen depletion, AMPK activation and consequent ACC inhibitory phosphorylation clearly occurred in GKO (Fig. 4E), suggesting that the intracellular glycogen in naïve ESCs is responsible for the constant inhibition of AMPK. GKO exhibited reduced intracellular fatty acid levels, as demonstrated by BODIPY staining (Fig. 4F). This suggests that ACC inhibition, triggered by AMPK activation (Fig. 4E) due to the depletion of glycogen (Fig. 4C), likely promotes fatty acid consumption (Fig. 4F). In line with this, a short-term withdrawal of glucose decreased the levels of fatty acids in GKO,

implying the crucial role of fatty acids as an energy reservoir in naïve ESCs (Fig. 4G). Additional assays were conducted to rule out the possibility of an unexpected off-target effect of KO, and similar results were obtained by stable knockdown (sh*Gys1*) in naïve ESCs (Fig. EV3C). The reduced glycogen (Fig. EV3D), AMPK activation and the concurrent ACC inhibition (Fig. EV3E) aligned with a lower fatty acid level (Fig. EV3F). AMPK remained active in *Gys1* knockdown cells even in the presence of iGSK3β, in contrast to the AMPK inhibition observed in control naïve ESCs (Fig. EV3G). These results clearly demonstrated that the intracellular glycogen produced by constant exposure to iGSK3β inhibited AMPK activation and contributed to the maintenance of intracellular fatty acid levels in naïve ESCs.

## Loss of glycogen primes exit of naïve pluripotency

Notably, the GKO displayed an aberrant colony, losing the characteristic dome-shape of naïve ESCs under 2i condition (Fig. 5A). The expression levels of typical naïve markers such as *Nanog, Rex1,* and *Esrrb* were notably reduced in GKO compared to control naïve ESCs (Fig. 5B). The subsequent transcriptome analysis revealed significant differences in gene clusters of GKO compared to the parent naïve and primed ESCs (Fig. 5C). Several gene sets associated with naïve (green) and primed (purple) pluripotency in GKO were positioned between the naïve and primed transcriptome signatures (Fig. 5D). We also observed the loss of the dome-shape colony morphology of control naïve ESCs at day 3 after glucose deprivation. Interestingly, this change resembled the colony morphology seen in GKO under normal culture condition (Fig. 5E). Prior to inducing cell death in GKO following glucose deprivation, a clear increase in expression of marker for primed pluripotency (*Fgf5*) and exit of pluripotency (*T*) as observed in GKO (Fig. 5F). To further highlight the effect of glycogen on the maintenance of naïve pluripotency, we employed an ESC line with two distinct fluorescence reporters under the control of the distal (DE) and proximal enhancer (PE) of *Pou5f1*. This system indicates naïve (expressing green fluorescence protein: GFP) and primed pluripotency (expressing red fluorescence protein: RFP), respectively (hereinafter referred to as OG-WT) (Choi et al, 2016). The state with simultaneous GFP and RFP expression, readily shift to GFP only with 2i or RFP only with bFGF2 and Activin A was defined as "metastable" as described previously (Choi et al, 2016). It is noteworthy that GFP positive, GFP/RFP double-positive and RFP positive ESCs in this model recapitulate the ICM of E3.5 blastocysts, E6.5 epiblasts and E7.5 epiblasts, respectively (Fig. 5G). This distinction capturing the pluripotency at different developmental stages, was used to examine the impact of glycogen on the

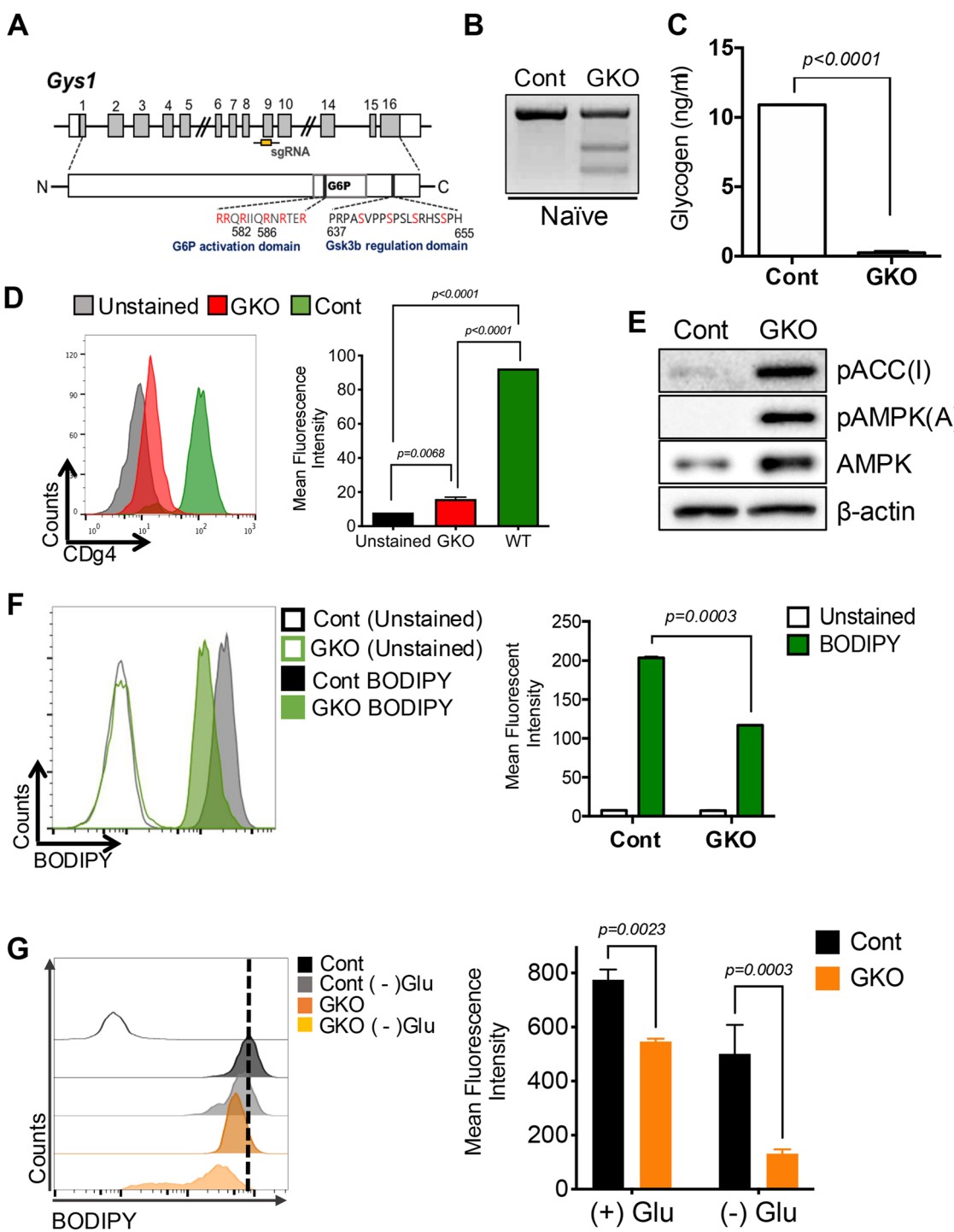

**Figure 4. Role of glycogen in the suppression of AMPK activation and maintaining fatty acid levels.**

(A) Scheme of *Gys1* KO by targeting exon 9. (B) T7E1 assay for WT and GKO in OG2$^{+/-}$GOF6$^{+/-}$ cell line. (C) Intracellular glycogen level of J1 Cont and GKO. Multiple t-tests, ($N = 3$, $n = 3$). This bar graph shows mean value with standard deviation as an error bar. (D) Flow Cytometry of CDg4 staining in J1 Cont and GKO (left panel), quantification of the mean fluorescence intensity of the flow cytometry (right panel). One-way ANOVA, multiple comparisons ($N = 3$, $n = 3$). This bar graph shows mean value with standard deviation as an error bar. (E) Immunoblotting analysis for indicative proteins (pAcc, pAMPKα, AMPK in J1 Cont and GKO) and β-actin is used for the loading control. (F) Flow cytometry of BODIPY 493/503 staining in J1 Cont and GKO (left panel), quantification of the mean fluorescence intensity of the flow cytometry (right panel), Multiple t-tests ($N = 3$, $n = 3$). This bar graph shows mean value with standard deviation as an error bar. (G) Flow cytometry of BODIPY 493/503 staining in J1 Cont and GKO with or without glucose (left panel), quantification of the mean fluorescence intensity of the flow cytometry (right panel). Two-way ANOVA, multiple comparisons, ($N = 3$, $n = 3$). This bar graph shows mean value with standard deviation as an error bar. Cont: Control, GKO: *Gys1* KO. Source data are available online for this figure.

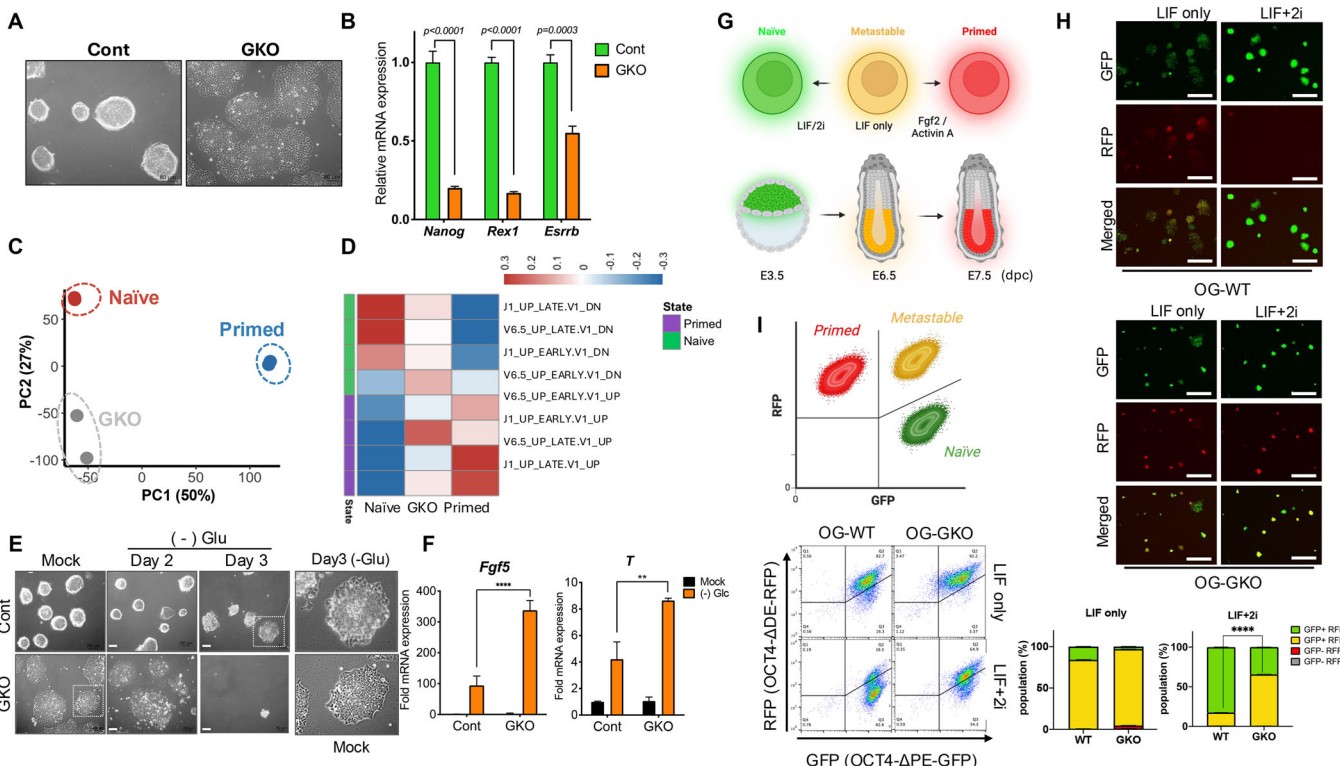

**Figure 5.  Loss of glycogen primes exit of naïve pluripotency.**

(**A**) Representative brightfield images of Cont and GKO mESCs. Scale bar: 80 μm. (**B**) Relative mRNA expressions for *Nanog*, *Rex1*, *Esrrb* in Cont and GKO. Multiple t-tests ($N = 3$, $n = 4$). This bar graph shows mean value with standard deviation as an error bar. (**C**) Principal component analysis (PCA) of the transcriptome of Naive (J1), GKO and Primed mESCs (PJI). (**D**) Gene set variation analysis (GSVA) of Naive and Primed signatures in Naive (J1), GKO and Primed mESCs (PJ1). (**E**) Representative brightfield image of Cont and GKO with (left) or without (right) Glucose. Scale bar, 200 μm. (**F**) Fold mRNA expressions of *Eomes* and *T* in Cont and GKO with [mock] or without [($-$) Glu] Glucose. Two-way ANOVA, (****$P < 0.0001$, **$P = 0.012$, $N = 3$, $n = 3$). This bar graph shows mean value with standard deviation as an error bar. (**G**) Graphical image of fluorescence activity of OG2$^{+/-}$GOF6$^{+/-}$ cell line in naïve, metastable, primed status (left panel), Graphical scheme of endogenous Oct4 allele, Oct4-ΔPE-GFP (OG2) allele, and Oct4-ΔDE-RFP (GOF6) allele (right panel). (**H**) Representative fluorescence images of OG-WT and OG-GKO (scale bas = 500 μm) under LIF only or LIF/2i conditions. (**I**) Flow cytometry of GFP and RFP in OG-WT and OG-GKO under LIF only or LIF/2i conditions (left panel), quantifications of the populations from the flow cytometry (right panel). Multiple t-tests (****$P < 0.0001$, $N = 3$, $n = 3$). This bar graph shows mean value with standard deviation as an error bar. Glu: Glucose. Cont: Control, GKO: *Gys1* KO. Source data are available online for this figure.

pluripotency as described previously (Kim et al, 2022c). Similar to the KO approach shown in Fig. 4A, another GKO line (OG-GKO) was established from the control OG-WT (Fig. EV5). When OG-WT and OG-GKO maintained at a "metastable" state with 'LIF only' were subjected to LIF/2i, OG-GKO still showed both GFP and RFP unlike the control ESCs, which were promptly converted to a GFP+ population (Fig. 5H). The consequent flow cytometry analysis highlighted the different reprogramming efficiency to naïve pluripotency from the metastable (GFP/RFP double positive) mESCs (Fig. 5I). These findings underscore the role of glycogen not only as an energy reservoir but also as a novel determinant for naïve pluripotency.

## AMPK is responsible for priming the exit of naïve pluripotency

As naïve-specific glycogen synthesis driven by GSK3β inhibition (Fig. 3) suppresses AMPK activation (Fig. 4) and supports the maintenance of naïve pluripotency (Fig. 5), we propose an intriguing hypothesis. Specifically, intracellular glycogen, which

gradually decreases from the blastocyst stage (Stern and Biggers, 1968) may act as a timer for determining the exit from naïve pluripotency by regulating AMPK activity and intracellular fatty acid levels (Fig. 6A). To test this hypothesis, we established AMPKα KO (AKO) and dual KO of *Gys1* and AMPKα (DKO) (Fig. 6B), as previously described (Zhang et al, 2019). The KO ESCs were readily isolated due to the simultaneous integration of RFP reporter through homologous directed repair (Fig. EV6A). Notably, we avoided the chemical perturbation of AMPK activity to eliminate potential unexpected side effects from prolonged exposure, as previously described (Liu et al, 2014). In the absence of AMPK in AKO and DKO, the inhibitory phosphorylation of ACC in GKO was markedly reduced in DKO (Fig. 6C), aligning with the recovery of reduced fatty acids of GKO (Fig. 6D) even when glycogen was lacking (Fig. 6E). The primed-like flat colony morphology in GKO as shown in Fig. 5A, was obviously reversed by the simultaneous KO of AMPK (i.e., DKO) (Fig. 6F), consistent to the levels of primed pluripotency markers (Fig. 6G). The hypothesis that timely AMPK activation occurring due to the progressive loss of glycogen at the blastocyst stage (i.e., naïve ESCs) instigates the epiblast

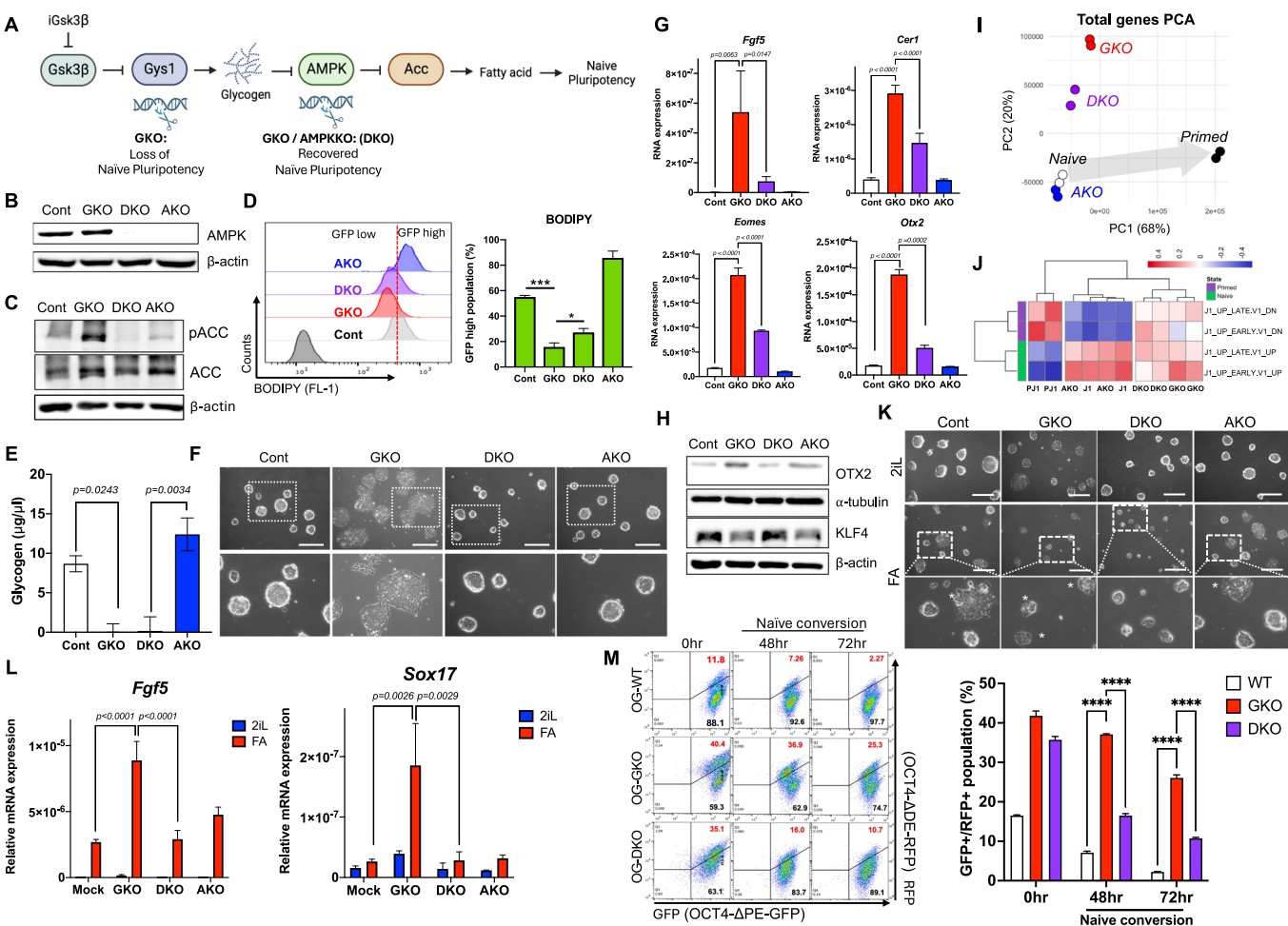

**Figure 6. AMPK is responsible for priming the exit of naïve pluripotency.**

(A) Graphical summary of GKO, AKO and DKO. (B) Immunoblotting analysis for indicative proteins (AMPK in Cont, GKO, DKO and AKO), β-actin was used as a loading control. (C) Immunoblotting analysis for indicative proteins (pAcc, ACC in Mock, GKO, DKO and AKO, β-actin was used as a loading control). (D) Flow cytometry of BODIPY 493/503 staining in Cont, GKO, DKO and AKO (left panel), quantification of the populations from the flow cytometry (right panel), Multiple t-tests (***$P = 0.0003$, *$P = 0.0243$, $N = 3$, $n = 3$). This bar graph shows mean value with standard deviation as an error bar. (E) Intracellular glycogen level of Cont, GKO, DKO and AKO was quantified. Two-way ANOVA, multiple comparisons ($N = 2$, $n = 3$). This bar graph shows mean value with standard deviation as an error bar. (F) Brightfield images of Cont, GKO, DKO, AKO (scale bar = 200 μm). Each inset is enlarged at the bottom of each condition, accordingly. (G) Relative mRNA expressions for *Fgf5, Cer1, Eomes* and *Otx2* in Cont, GKO, DKO and AKO under LIF/2i media condition. Two-way ANOVA, multiple comparisons (all panels; $N = 3$, $n = 3$). This bar graph shows mean value with standard deviation as an error bar. (H) Immunoblotting analysis for indicative proteins (Otx2, Klf4 in Cont, GKO, DKO and AKO), β-actin and α-tubulin were used as loading controls. (I) PCA of RNA-seq data from Naive (J1), AKO, DKO, GKO and primed (PJ1). The gray arrow indicates a suggestive transition from naïve-to-primed pluripotency. (J) Gene set variation analysis (GSVA) of Naive and Primed signatures from Naive (J1), AKO, DKO, GKO and Primed mESCs (PJ1). (K) Brightfield images of Cont, GKO, DKO, AKO under indicative media conditions. Primed conversion was performed by adding KOSR, bFGF and Activin A for 48 h. Each inset in upper images is enlarged at bottom accordingly (scale bar = 200 μm). Asterisks indicates the typical morphology of flat colony representing exit of naïve pluripotency. (L) Relative mRNA expressions of *Fgf5* and *Sox17* under indicative media conditions (2iL and FA). Primed conversion was performed by adding KOSR, bFGF and Activin A for 48 h. Two-way ANOVA, multiple comparisons ($N = 3$, $n = 3$). This bar graph shows mean value with standard deviation as an error bar. Cont: Control, GKO: *Gys1* KO, AKO: *Prkaa1* KO, DKO: *Gys1* KO and *Prkaa1* KO, KOSR: Knock Out Serum Replacement, 2iL: 2i + LIF, FA: bFGF + Activin A. (M) Flow cytometric analysis to examine GFP+ and RFP+ population in OG-WT, OG-GKO and OG-DKO mESCs during the naïve conversion (serum-free 2i + LIF) (left panel), and both GFP+ and RFP+ expressing population (GFP + /RFP + ) is quantified in each line (right panel). Two-way ANOVA, multiple comparisons (****$P < 0.0001$, $N = 3$, $n = 5$). This bar graph shows mean value with standard deviation as an error bar. Source data are available online for this figure.

transition (i.e., primed ESCs), was also supported by key findings. *Otx2*, one of the intrinsic determinants of epiblast transition (Acampora et al, 2013), was upregulated both at the transcription (Fig. 6G) and protein level (Figs. 6H and EV6B). This upregulation showed an inverse correlation with KLF4, a factor solely sufficient to induce the transition from epiblast stem cells to naïve cells (Guo et al, 2009). *Otx2* induction, driven by reduced Wnt signaling in

blastocysts, promotes the transition to the rosette state (Neagu et al, 2020). This prompted us to determine and compare the transcriptome signatures of naïve, AKO, GKO, DKO, and primed ESCs (bulk RNAseq data are available at GSE272593). Consistent with the observed colony morphology (Fig. 6F) and marker gene expression (Fig. 6G), the altered transcriptome profile of GKO was partially reversed in DKO, while AKO remained similar to naïve

ESCs (Fig. 6H–J; Dataset EV1). Furthermore, a meta-analysis of transcriptomes from typical intermediate states between naïve and primed pluripotency—including rosette (GSE105762), formative (GSE131556), and diapause (GSE143494)—was compared to that of GKO. As predicted by Otx2 induction, the transcriptome profile of GKO closely resembled that of the rosette state (Figs. EV6C and 6D) but differed from the formative (Fig. EV6E) and diapause (Fig. EV6F) states. Next, these ESCs were subjected to a primed transition with Fgf2/Activin A. After 48 h of exposure to Fgf2/Activin A, a predominance of flat-like colony morphology observed in GKO was reversed in DKO (Fig. 6K). These morphological observations were corroborated by the expression levels of typical primed pluripotency markers (e.g., *Fgf5* and *Sox17*) (Fig. 6L). Beyond morphology and marker expression, our findings were further substantiated by the dual reporter system. We established an additional KO of AMPKα from OG-GKO (referred as OG-DKO) (Fig. EV6G). The functional loss of AMPKα was evident through AMPK and ACC phosphorylation (Fig. EV6H), consistent morphological differences (Fig. EV6I) and the expression levels of key markers for naïve pluripotency (Fig. EV6J). Monitoring the naïve transition up to 72 h, clearly demonstrated that OG-DKO showed recovered naïve conversion efficiency from that of OG-GKO, suggesting the loss of AMPK activity may play a critical role in naïve reprogramming in mESCs (Fig. 6M). In summary, these results collectively suggest that AMPK activation triggered by the gradual expenditure, acts as a switch for the exit from naïve pluripotency during the peri and post-implantation process.

## Discussion

The limited availability of in vitro models for studying mammalian embryos has posed challenges in the biological characterization of pre- and post-implantation embryos. Recent advancements, such as the development of blastoids (blastocyst-like structure) (Rivron et al, 2018), and other ex-vivo embryo-like structure, have aimed to simulate mouse and human embryo development (Bao et al, 2022). Nevertheless, the use of naïve and primed ESCs, enabling extensive biochemical and molecular studies, remains crucial for understanding pluripotency regulatory mechanisms in pre- and post-implantation embryos, respectively (Nichols and Smith, 2009; Nichols and Smith, 2012; Ying et al, 2008).

This study emphasizes the significance of glycogen as a signaling molecule in regulating AMPK activation during early embryonic development. While glycogen has traditionally been viewed as an energy reservoir, our results provide evidence that glycogen levels in blastocysts can serve as a regulatory mechanism for modulating AMPK activity. The observed decrease in glycogen levels in mouse blastocysts prior to implantation (Ozias and Stern, 1973) suggests that timely AMPK activation through glycogen expenditure may occur in vivo during the pre-implantation stage of embryo development. The critical role of AMPK activation in early embryogenesis is supported by genetic studies showing embryonic lethality in AMPKα1/α2 KO mice around 10.5 days post-conception (Viollet et al, 2009) and the failure of differentiation in AMPKα/β KO ESCs (Young et al, 2016). In this line, sustained GSK3α/β inhibition by Netrin-1 signaling (Huyghe et al, 2020) would likely lead to AMPK activation when glycogen is depleted,

whether through Netrin-1 signaling in embryos or a chemical inhibitor of GSK3β in naïve ESCs.

Fatty acid metabolism, particularly fatty acid β-oxidation (FAO), has been extensively studied in reproductive biology and embryonic development (Dunning et al, 2010). Key genes involved in fatty acid synthesis such as *Fasn* (Chirala et al, 2003) and *ACC* (Abu-Elheiga et al, 2005) have been shown to be essential for early embryo development, which is consistent with our observation that chemical inhibition of ACC can induce cell death in naïve ESCs (Fig. 2I). Consistently, treatment with L-carnitine, which promotes FAO and potentially decreases fatty acid levels in embryos, improves pregnancy rates following embryo transfer in bovine models (Carrillo-Gonzalez and Maldonado-Estrada, 2020) and humans (Kim et al, 2018). Interestingly, mammalian species with higher levels of fatty acid in embryos such as cows, pigs, and cats (Krisher and Prather, 2012) exhibit longer implantation times compared to mice, which have lower levels of fatty acid in embryos (Simmet et al, 2018). This implies that fatty acid metabolism, regulated by AMPK activity, may play a pivotal role in determining the timing of implantation across different species. Another intriguing question to be addressed is the molecular mechanism how fatty acids delay the exit of naïve pluripotency. Previous studies have indicated that fatty acid metabolism, including FAO, provides acetyl carbon for histone acetylation (McDonnell et al, 2016) and is critical for providing a carbon source for pluripotency maintenance (Khoa et al, 2020). This interesting research subjects would be investigated in future studies.

Collectively, the findings of this study propose that glycogen stored in blastocysts acts as a signaling molecule, regulating the timely activation of AMPK. Consequently, this regulation influences the levels of fatty acids, ultimately initiating the departure from naïve pluripotency. These observations may offer a potential explanation for the variations in fatty acid levels seen in embryos across diverse animal species, each characterized by distinct implantation periods. Overall, these results contribute novel insights into the role of glycogen as a signaling molecule in early embryonic development and provide illumination on the molecular mechanisms that govern the regulation of pluripotency and implantation.

## Methods

**Reagents and tools table**

| Reagent/resource | Reference or source | Identifier or catalog number |
|---|---|---|
| **Experimental models** | | |
| OG2GOF6 mESC | Choi et al, 2016 | |
| J1 mESC | ATCC | SCRC-1010 |
| NIH3T3 | ATCC | CRL-1658 |
| **Recombinant DNA** | | |
| pRGEN-Cas9-CMV-Puro-RFP | ToolGen | TGEN-OP1 |
| pRG2 | Addgene | 104174 |
| Tet-pLKO-Puro | Addgene | 21915 |

| Reagent/resource | Reference or source | Identifier or catalog number |
|---|---|---|
| AMPK alpha 1 CRISPR Plasmids | Santa Cruz | sc-430618 |
| AMPK alpha 1 HDR Plasmids | Santa Cruz | sc-430618-HDR |
| **Antibodies** | | |
| Rabbit anti-AMPKα | Cell Signaling Technology | 2532 |
| Rabbit anti-pAMPKα (T172) | Cell Signaling Technology | 2535 |
| Rabbit anti-ACC | Cell Signaling Technology | 3676 |
| Rabbit anti-pACC (S79) | Cell Signaling Technology | 11818 |
| Rabbit anti-pGSK3β(S9) | Cell Signaling Technology | 9323 |
| Rabbit anti-pGSK3β(Y216) | Abcam | ab75745 |
| Rabbit anti-GYS1 | Cell Signaling Technology | 3886 |
| Rabbit anti-pGYS1 (S641) | Cell Signaling Technology | 47043 |
| Rabbit anti-pSTAT3 (Y705) | Cell Signaling Technology | 9145 |
| Rabbit anti-pERK1/2 (T202/Y204) | Cell Signaling Technology | 9101 |
| Mouse anti-β-actin | Santa Cruz | sc-47778 |
| OCT4 | Cell Signaling Technology | 2840 |
| KLF4 | Abcam | ab72543 |
| Mouse anti-β-α-tubulin | Santa Cruz | sc-8035 |
| OTX2 | Abcam | ab183951 |
| Vinculin | Santa Cruz | sc-25336 |
| Peroxidase AffiniPure™ Goat anti-mouse IgG (H + L) | Jackson ImmunoResearch Lab | 115-035-003 |
| Peroxidase AffiniPure™ Goat anti-rabbit IgG (H + L) | Jackson ImmunoResearch Lab | 111-035-003 |

| **Oligonucleotides and other sequence-based reagents** | **Sequence (5′–3′)** | |
|---|---|---|
| sgGys1 | GGACACAGCCAATACAGTCA | |
| sgPrkaa1 | GCCGCACCAGAAGTCATTTC | |
| shGys1 | CAAGGGTTGTAAGGTGTATTT | |
| Esrrb | F: GATTCTCATCTTGGGCATCGTGTAC R: CTGACTCAGCTCATAGTCCTGCAG | |
| Klf2 | F: CACACATACTTGCAGCTACACCAAC R: CAAGTGGCACTGAAAGGGTCTGTG | |

| Reagent/resource | Reference or source | Identifier or catalog number |
|---|---|---|
| Rex1 | F: CTTCGAAAGCTTGGAGGAAGTGGAG R: GGACACTCCAGCATCGATAAGACAC | |
| Klf4 | F: GAACAGCCACCCACACTTGTGAC R: CTGTCACACTTCTGGCACTGAAAG | |
| NrOb1 | F: ACAGAGCAGCCACAGATGGTGTC R: GATGTGCTCAGTAAGGATCTGCTG | |
| Gys1 | F: AACAAGGTGGGTGGCATCTA R: CCTTACAACCCTTGCTGTTC | |
| Nanog | F: GTGCACTCAAGGACAGGTTTCAG R: CTGCAATGGATGCTGGGATACTC | |
| Fgf5 | F: CATCGGTTTCCATCTGCAGATCTAC R: GTTCTGTGGATCGCGGACGCATAG | |
| Sox17 | F: ACCCAGATCTGCACAACGCAGAG R: GCTTCATGCGCTTCACCTGCTTG | |
| T | F: CATCTGCTTGTCTGTCCATGCTG R: GAGAACCAGAAGACGAGGACGTG | |
| Cer1 | F: GTGGAAAGCGATCATGTCTCATCG R: GCAAAGGTTGTTCTGGACAACGAC | |
| Eomes | F: CTCAGAGACACAGTTCATCGCTGTG R: CAGGGACAATCTGATGGGATCTAGG | |
| Otx2 | F: TCATGAGGGAAGAGGTGGCACTG R: AGCACTGCTGCTGGCAATGGTTG | |
| Rn18s | F: GTAACCCGTTGAACCCCATT R: CCATCCAATCGGTAGTAGCG | |
| **Chemicals, enzymes and other reagents** | | |
| DMEM high glucose | Gibco | 11965092 |
| DMEM/F12 | Gibco | 21331020 |
| N2 supplement | Gibco | 17502048 |
| B27 supplement | Gibco | A3582801 |
| Fetal Bovine Serum (FBS) | Gibco | 16000044 |
| Knockout Serum Replacement | Gibco | 10828028 |
| DPBS | Welgene | LB001-02 |
| MEM-nonessential amino acids | Gibco | 11130051 |

| Reagent/resource | Reference or source | Identifier or catalog number |
|---|---|---|
| GlutaMax | Gibco | 35050061 |
| Gentamycin | Gibco | 15710064 |
| β-mercaptoethanol | Gibco | 21985023 |
| mouse leukemia inhibitory factor (mLIF) | Merck Millipore | ESG1107 |
| Porcine Gelatin | Merck Millipore | G1890 |
| 0.25% Trypsin/EDTA solution | Welgene | LS015-10 |
| Accutase | BD-Bioscience | 561527 |
| Dispase | Gibco | 17105041 |
| Matrigel | Corning | 354277 |
| PD0325901 | Biogems | 3911091 |
| CHIR99021 | Biogems | 2520691 |
| PF-05175157 | MedChemExpress | HY-12942 |
| Murine Activin-A | Peprotech | 120-14E |
| Murine bFgf | Peprotech | 450-33 |
| G418 | Merck Millipore | G8168 |
| BODIPY493/503 | Invitrogen | D3922 |
| Penicillin-Streptomycin-GlutaMAX | Gibco | A5873601 |
| T7 Endonuclease I | New England Biolabs | M0320S |
| RIPA buffer | Biosesang | R2002-050-00 |
| 5× SDS-PAGE loading buffer | Biosesang | SF2088-110-00 |
| Protease Inhibitor Cocktail | Merck Millipore | C756V54 |
| Miracle-Star | iNtRON Biotechnology | 16028 |
| West-Queen | iNtRON Biotechnology | 16026 |
| ECL Select | Cytiva | RPN2235 |
| 5× PrimeScript™ RT mix | Takara | RR036A |
| 2× TB-Green premix | Takara | RR820S |
| CDg4 | Lee et al, 2012 | |
| Nile Red | Merck Millipore | 72485 |
| $^{13}$C-glucose | Cambridge Isotope Laboratories | TRC-G595008 |
| Chloroform-d$_6$ | Merck Millipore | 151823 |
| CPTCI CryoProbe | Bruker BioSpin | |
| Easy RNA Directional Library Prep Kit | MGI | 1000006386 |
| QantiFluor ONE dsDNA System | Promega | E4871 |
| QantiFluor ssDNA System | Promega | E3190 |
| Propidium Iodide | Merck Millipore | 537059 |
| 7-AAD | BD Pharmingen | 559925 |

| Reagent/resource | Reference or source | Identifier or catalog number |
|---|---|---|
| Hoechst 33342 | Invitrogen | H1399 |
| Lipofectamine-3000 | Invitrogen | L3000-001 |
| **Software** | | |
| FlowJo | https://www.bdbiosciences.com/en-us/products/software/flowjo-v10-software | |
| GraphPad Prism 9 | https://www.graphpad.com/features | |
| Fiji | https://imagej.net/software/fiji/downloads | |
| R version 4.2.3 | https://www.r-project.org/ | |
| FASTQC version 0.11.9 | https://www.bioinformatics.babraham.ac.uk/projects/fastqc/ | |
| TrimGalore version 0.6.6 | https://www.bioinformatics.babraham.ac.uk/projects/trim_galore/ | |
| STAR version 2.7.9a | https://github.com/alexdobin/STAR | |
| RSEM version 1.3.3 | https://github.com/deweylab/RSEM | |
| Gene Expression Omnibus (GEO) | https://www.ncbi.nlm.nih.gov/geo/ | |
| Gene Set Enrichment Analysis (GSEA) | https://www.gsea-msigdb.org/gsea/index.jsp | |
| PhosphoSitePlus | https://www.phosphosite.org/homeAction.action | |
| **Other** | | |
| Wizard genomic DNA isolation kit | Promega | A1120 |
| Easy-Blue™ total RNA isolation kit | iNtRON Biotechnology | 17061 |
| Annexin-V Apoptosis Detection Kit | BD Pharmingen | 559763 |
| Pierce BCA protein assay Kit | Thermo Fischer | 23225 |
| Glycogen Assay Kit | BioVision | K646-100 |

## Cell culture

naïve mouse ESCs were cultured on 0.5% porcine gelatin-coated dish in either naïve mESC culture media -DMEM high glucose (Gibco) supplemented with 15% FBS (Gibco), 1% Glutamax (Gibco), 1% MEM-nonessential amino acids (Gibco), 0.1% Gentamycin (Gibco), 0.1 mM β-mercaptoethanol (Gibco), 1000 U/ml mouse leukemia inhibitory factor (mLIF) (Millipore, Merck), 1 μM PD0325901 (Peprotech) and 3 μM CHIR99021—or N2B27 naïve mESC culture media—a 1:1 ratio of DMEM/F12 (Gibco) and Neurobasal (Gibco) supplemented with 1% P/S/G (Gibco), 0.5% N2 (Gibco) and 1% B27 (Gibco), 1000 U/ml mouse leukemia inhibitory factor (mLIF) (Millipore, Merck), 1 μM PD0325901 (Peprotech) and 3 μM CHIR99021—at 37 °C and 5% $CO_2$ incubating condition. Cells were passaged 1:20 ratio every 3 days using 0.25% Trypsin/EDTA (Welgene) as a single cell dissociation reagent. OG2 + /−GOF6 + /− cells were cultured on 0.5% porcine gelatin-coated dish in naïve mESC culture media with or without 2i (1 μM PD0325901

(Peprotech) and 3 µM CHIR99021). Primed mouse ESCs were cultured on Matrigel (Corning# 354277)-coated dish in either EpiSC culture media—DMEM/F12 (Gibco) supplemented with 20% KnockOut Serum Replacement (Gibco), 1% GlutaMAX (Gibco), 1% MEM-nonessential amino acids (Gibco), 0.1% Gentamycin (Gibco), 10 ng/ml murine bFgf (Peprotech), 20 ng/ml murine Activin A (Peprotech)—or N2B27 EpiLC culture media—a 1:1 ratio of DMEM/F12 (Gibco) and Neurobasal (Gibco) supplemented with 1% P/S/G (Gibco), 0.5% N2 (Gibco) and 1% B27 (Gibco), 1% KnockOut Serum Replacement (Gibco), 12 ng/ml murine bFgf (Peprotech), 20 ng/ml murine Activin A (Peprotech)—at 37 °C and 5% $CO_2$ condition. Primed mESCs were passaged every 3 days using 1 U/ml of Dispase (Gibco) as a colony detachment reagent. Detached colony clumps were transferred 1:15–1:20 ratio on Matrigel (Corning# 354277) coated dishes. Culture media was changed every day for all cell types.

## Establishment and sequence validation of knock out cell lines

For establishment of Gys1 KO mESCs, we transfected pRGEN-Cas9-CMV-Puro-RFP (Toolgen-TGEN_OP1) plasmids with sgRNA of Gys1 cloned into pRG2 (Addgene# 104174) plasmid with following sgRNA sequence (5'-GGACACAGCCAATA-CAGTCA-3') as described previously (Park et al, 2023). For establishment of Prkaa1 KO OG2 + /− GOF6 + /− cell line, we transfected pRGEN-Cas9-CMV-Puro-RFP (Toolgen-TGEN_OP1) plasmids with sgRNA of Prkaa1 cloned into pRG2 (Addgene# 104174) plasmid with following sgRNA sequence (5'-GCCGCAC-CAGAAGTCATTTC-3'). The cloned gRNA vector (3 mg) and Cas9 plasmid vector (1 mg) were co-transfected into $1 \times 10^6$ mESCs using Lipofectamine 3000 reagent (#L3000-001, Invitrogen). After 24 h, cells were selected with puromycin (2 µg/mL) for 24 h. Single-colony picking was performed from KO pool. Targeted sequence of each single clone was validated by Sanger sequencing after gDNA isolation through Wizard® Genomic DNA Purification Kit (#A1120, Promega). For establishment of Prkaa1 KO J1 cell line, commercial CRISPR/Cas9 Knockout Plasmid (#sc-430618, Santa Cruz) and HDR plasmid (#sc-430618, Santa Cruz) were co-transfected using Lipofectamine 3000 reagent (#L3000-001, Invitrogen). After 24 h, RFP positive cells were sorted using FACS Aria III cell sorter (BD Biosciences). After both genetic and functional validation of pooled clones, we took untargeted clone as "Cont" and gene-of-interest targeted clones for "KO" for further experiments.

## T7E1 assay

KO pool cells were collected and gDNA was extracted using Wizard® Genomic DNA Purification Kit (#A1120, Promega) following the manufacturer's instruction. T7E1 assay was performed as previously described (Kim et al, 2020).

## Establishment of knock down cell lines

For mouse shGys1 plasmid vector generation, we used PB-PGK-Neo empty vector as a backbone vector and the shRNA sequence for mouse Gys1 was infusion cloned into the plasmid vector (shGys1 sequence, forward: 5'-CAAGGGTTGTAAGGTGTATTTC TCGAG-3', reverse: 5'- CAAGGGTTGTAAGGTGTATTTCTCGA G-3'). To establish Gys1 stable knock-down cell line, 2 mg of shGys1 Piggy-Bac vector and 1 mg of Transposase vectors was co-transfected into $1 \times 10^6$ cells of mESCs through Lipofectamine-3000 transfection (#L3000-001, Invitrogen). After 24 h, cells were treated with 100 µg/mL of G418 (Sigma) for 2 days followed by washing-off G418 on following day.

## Immunoblotting analysis

RIPA buffer (Biosesang) containing 1 µM protease inhibitor and 10 µM sodium orthovanadate was used to extract the whole cell lysate, which was then acquired after incubating on ice for 1 h and subsequent centrifugation. Quantification of proteins was performed using the Pierce BCA protein assay Kit (Thermo Fischer Scientific). To prepare the protein sample, 5× SDS-PAGE loading buffer (Biosesang) was added, and the sample was boiled at 100 °C for 10 min. In total, 10–20 µg of total protein was loaded and separated on a 10% SDS-PAGE gel. The separated proteins were transferred onto an activated PVDF membrane. The membrane with transferred proteins was blocked with 5% skim milk in TBS-T at RT for 1 h, followed by washing. The primary antibody (1:500–1:1000) in TBS-T was incubated with 1% sodium azide at 4 °C overnight. After washing, the membrane was incubated with the secondary antibody (1:10,000) in TBS-T at RT for 1 h. Chemiluminescence was detected using the Miracle-Star (iNtRON Biotechnology) kit or West-Queen (iNtRON Biotechnology) kit. The band quantification was performed using Image J software (Fiji).

## RNA isolation and quantitative RT-PCR analysis

The Easy-Blue™ total RNA isolation kit (iNtRON Biotechnology) was used to isolate total RNA from cells, following the manufacturer's instructions. During reverse transcription, 5× PrimeScript™ RT mix (TaKaRa) was used to acquire cDNA. Quantitative real-time PCR was carried out using 2× TB-Green premix (TaKaRa) on a LightCycler-480II (Roche). The Rn18s gene was used as an internal loading control to normalize the gene expression data.

## CDg4 and Nile Red staining and quantification

For CDg4 and Nile Red staining of in vitro cultured cells, cells were dissociated with Accutase (#561527, BD Biosciences) and washed with DPBS. 1 million cells were counted followed by fixation with 4% PFA at RT for 5 min. After fixation, cells were incubated with CDg4 dye (3 µM) or 1 µg/mL of Nile Red in DPBS at 37 °C for 1 h. After washing with DPBS, each fluorescent dye was analyzed by flow cytometry. For blastocysts staining, 3.5–4.5dpc mouse embryos were collected fixed with 4% PFA followed by 3 µM of CDg4 or 1 µg/mL of Nile Red staining for 1 h at 37 °C. In all, 10 µg/mL of Hoechst was counter stained to indicate nucleus of each embryo. The fluorescent and bright field images of the embryo were taken after the staining.

## BODIPY 493/503 staining and quantification

For BODIPY 493/503 staining of in vitro cultured cells, cells were dissociated with Accutase (#561527, BD Biosciences) and washed with DPBS. 1 million cells were counted and incubated with BODIPY 493/503 (2 µM) in DPBS at 37 °C for 30 min. After washing with DPBS, BODIPY 493/503 staining was analyzed by flow cytometry.

## Propidium iodide (PI) staining for cell cycle analysis

For PI staining of in vitro cultured cells, cells were dissociated with Accutase (#561527, BD Biosciences) and washed with DPBS. One million cells were counted followed by fixation with 75% EtOH at RT for 15 min. Cells were then incubated with 1% RNase A in DPBS at RT for 30 min, followed by staining with 10% PI solution in DPBS. After washing with DPBS, PI staining was analyzed by flow cytometry.

## Flow cytometric analysis

Flow cytometric analysis was used to measure the GFP intensity (following BODIPY 493/503 and CDg4 staining), cell cycle profile (following PI staining), and GFP and RFP intensity in the OG2GOF6 cell line. Cells were dissociated with Accutase (#561527, BD Biosciences) and washed with DPBS, and stained as needed. Cells were analyzed through FACS Calibur (BD Biosciences) or FACS Celesta (BD Biosciences) flow cytometer. GFP and CDg4 intensity was determined by measuring the FL-1 channel (FACS Calibur) or FITC (FACS Celesta). PI staining intensity was measured in the FL-2 channel (FACS Calibur), and RFP and Nile Red intensity were measured in the FL-3 channel (FACS Calibur). FlowJo software was used to analyze the flow cytometric data.

## NMR analysis with $^{13}$C-glucose tracer

Cells were cultured with 15 mM $^{13}$C-glucose (U-$^{13}C_6$, 99%, Cambridge Isotope Laboratories) for 24 h. After harvesting, cell pellets underwent standard two-phase extraction, and the chloroform phase underwent speedvac. The dried samples were dissolved in chloroform-$d_6$ (Sigma-Aldrich) and $^{1}$H-$^{13}$C Heteronuclear Single Quantum Coherence (HSQC) NMR was taken, using 800 MHz Bruker Avance III HD spectrometer equipped with a 5 mm CPTCI CryoProbe (Bruker BioSpin, Germany). The time domain parameter for carbon was 512, the spectral width for carbon was 80, and the number of scans was 3.

## Cell imaging

Brightfield images of live cells were captured by Olympus CKX41. GFP and RFP images of live cells were captured by JuLi Stage (NanoEntek), followed by the merging process of each channel by JuLi-Edit software.

## Glycogen quantification assay

Glycogen assay kit (BioVision) was used to measure the relative intracellular glycogen amount. For the sample preparation, cells were dissociated with Accutase (#561527, BD Biosciences) and washed with DPBS. 3 million cells were homogenized with cold distilled water on ice for 1 h. Supernatant was collected after spin down (13,000 rpm, 20 min). BCA assay was performed with the lysate. Then the lysate was boiled at 100 °C for 10 min. In total, 30–50 μg of protein was loaded to each well of 96-well plate and glycogen assay was performed under the manufacturer's instruction. The optical density of glucose (which was hydrolyzed from glycogen) was measured at 570 nm by Epoch microplate spectrophotometer (Biotek). Final glycogen concentration was normalized into initial protein input of each sample.

## Bulk RNA-seq library preparation

Total RNA was isolated from J1, GKO and PJ1 cells using Easy-BLUE™ RNA isolation kit (iNtRON Biotechnology, #17061). One 1 μg of total RNA was processed for preparing mRNA sequencing library using MGIEasy RNA Directional Library Prep Kit (MGI) according to manufacturer's instruction. The first step entails utilizing poly-T oligo-attached magnetic beads to isolate the mRNA molecules that contain poly-A. Following purification, divalent cations and a high temperature are used to break the mRNA into small pieces. Utilizing reverse transcriptase and random primers, the cleaved RNA fragments are converted into first strand cDNA. After achieving strand specificity in the RT directional buffer, second strand cDNA synthesis takes place. The 'A' base is then added to these cDNA fragments, followed by the ligation of the adapter. The final cDNA library is made by purifying and enriching the results with PCR. The QauntiFluor ONE dsDNA System (Promega) is used to quantify the double stranded library. The library is circularized at 37 °C for 30 min and then digested at 37 °C for 30 min, followed by cleanup of circularization product. The library is treated with the DNB enzyme at 30 °C for 25 min to create DNA nanoballs (DNB). Finally, Library was quantified by QauntiFluor ssDNA System (Promega). On the MGIseq system (MGI), the prepared DNB was sequenced using 100 bp paired-end reads.

## Bulk RNA-seq processing and analysis

Low-quality bases and adapter sequences bases were trimmed using TrimGalore (https://www.bioinformatics.babraham.ac.uk/). The trimmed reads were aligned to the mouse genome assembly GRCm39 using STAR (v2.7.3a). The expression value per gene was estimated as a read count or transcripts per million (TPM) values calculated using RSEM (v1.3.3) based on the mouse gene annotation GRCm39.104. 21,853 protein-coding genes were utilized for subsequent analysis. The fastq files and pre-processed data are available in the Gene Expression Omnibus (https://www.ncbi.nlm.nih.gov/geo/) under accession number GSE272593. To assign pathway activity scores for individual samples (naïve (J1), AKO, DKO, GKO and Primed (PJ1)), the single-sample GSEA (ssGSEA), using a profile of TPM values for all genes as an input, was conducted through the R package gsva.

## Transcriptome data analysis

To compare the generated RNA-seq data with existing dataset, RNA-seq data representing the rosette status (Alex Neagu et al, 2020), formative status (PMID: 34861148) and Diapause status (PMID: 31991105) were utilized. All datasets, including the newly generated data, were processed in a consistent manner to ensure compatibility. The EdgeR package was employed to identify genes associated with rosette and diapause statuses, while the DESeq2 package was used to identify genes specific to the formative status. Principal Component Analysis (PCA) was performed on both the publicly available and newly generated RNA-seq datasets to extract the first and second principal components (PC1 and PC2).

## Phosphoproteome data analysis

The processed data from Ana Martinez-Val et al's study was analyzed using a two-sided limma approach (version 3.54.2) to determine differential phosphorylation sites at each time point compared to the initial Serum/LIF conditions (Martinez-Val et al, 2021); Data ref: Martinez-Val et al, 2021). The statistical threshold was adjusted using the Benjamini–Hochberg correction to control the false discovery rate (FDR) at 5% for each condition.

## Phosphoprotein clustering based on inhibitors-induced phosphorylation changes

To group phosphoproteins based on their phosphorylation changes in response to inhibitors, including iMEK1, iGSK3β, and iCdk8, we employed hierarchical clustering with a predefined cluster number of $K = 4$. In this clustering process, phosphorylation values were first normalized across all samples using z-transformation, ensuring that the values were on a comparable scale. Pairwise distances between phosphoproteins were then computed using the Euclidean distance metric, based on the mean phosphorylation values across all samples. The hierarchical clustering algorithm was applied to the resulting distance matrix using the complete linkage method. To identify whether specific clusters were enriched with substrates downstream of particular kinases, we conducted a hypergeometric test using the PhosphoSitePlus database. This analysis allowed us to determine which clusters contained a significant number of substrates regulated by specific kinases.

## Statistical analysis

The mean values of the quantitative data were presented with their corresponding standard deviation (SD). To determine the statistical significance of each response variable, unpaired two-tailed t-tests were performed. Where necessary, pre-specified comparisons between groups were conducted using Tukey's post hoc test in PRISM software. P-values less than 0.05 were considered statistically significant (* < 0.05, **<0.01, ***<0.001, ****<0.0001 and n.s. for not significant).

# Data availability

The transcriptome datasets for naïve, AKO, GKO, DKO, and primed ESCs are deposited at GSE272593. Additional datasets used and/or analyzed during this study are available from the corresponding author upon reasonable request.

The source data of this paper are collected in the following database record: biostudies:S-SCDT-10_1038-S44319-025-00384-x.

# Peer review information

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

## Acknowledgements

This work was supported by a grant from Ministry of Science and ICT, and Ministry of Health and Welfare (Grant number RS-2024-00432867 and RS-2023-00218543), Republic of Korea. The authors also greatly appreciate Chi TN Nguyen and YP Kang from Seoul National University, College of Pharmacy for their metabolome analysis during the revision process of this study.

## Author contributions

**Seong-Min Kim**: Data curation; Writing—original draft. **Eun-Ji Kwon**: Data curation; Visualization; Writing—original draft. **Ji-Young Oh**: Data curation. **Han Sun Kim**: Data curation. **Sunghyouk Park**: Data curation. **Goo Jang**: Resources. **Jeong Tae Do**: Resources. **Keun-Tae Kim**: Conceptualization; Data curation; Supervision; Writing—original draft; Writing—review and editing. **Hyuk-Jin Cha**: Conceptualization; Supervision; Funding acquisition; Writing—original draft; Writing—review and editing.

Source data underlying figure panels in this paper may have individual authorship assigned. Where available, figure panel/source data authorship is listed in the following database record: biostudies:S-SCDT-10_1038-S44319-025-00384-x.

## Disclosure and competing interests statement

The authors declare no competing interests.

# Expanded View Figures

**Figure EV1.   Phosphoproteomic analysis of temporal kinase activity.**

(**A**) Volcano plot of differentially phosphorylated proteins after 1, 2 and 6 h of treatment ($n = 4$) compared to SL ($n = 8$). We employed moderated t-test which were implemented in the R package limma (v3.62.1). Differentially phosphorylated proteins were selected based on a Benjamini–Hochberg-adjusted P-value less than 0.05 and an absolute fold-change greater than 0 as the cut-offs. (**B**) Phosphoproteins phosphorylated by MAPK1, ERK1 and GSk3 among the phosphoproteins belonging to cluster 3. (**C**) PCA of phosphoproteome data from 2i treatment (0.5-, 1-, 2-, and 6-h samples displayed in red gradient color) and Cdk8i treatment (0.5, 1, 2, and 6-h samples displayed in green gradient color). (**D**) Hierarchical clustering results representing four phosphoproteins' clusters showing distinct phosphoproteome changes at various time points after CDK8i treatment. The top five kinase categories significantly enriched in each phosphoprotein cluster. (**E**) Phosphoproteins phosphorylated by AMPK among the phosphoproteins belonging to cluster 1.

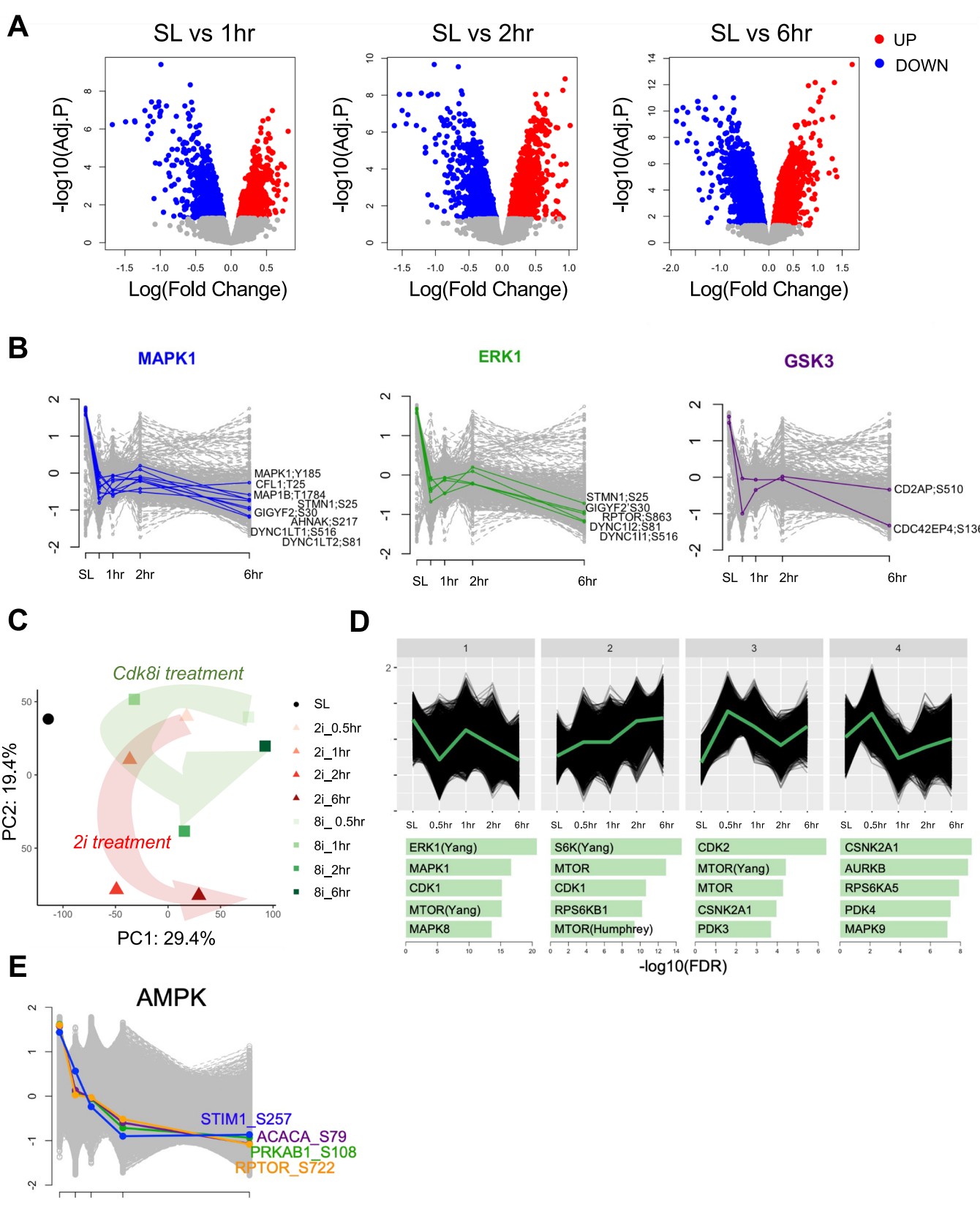

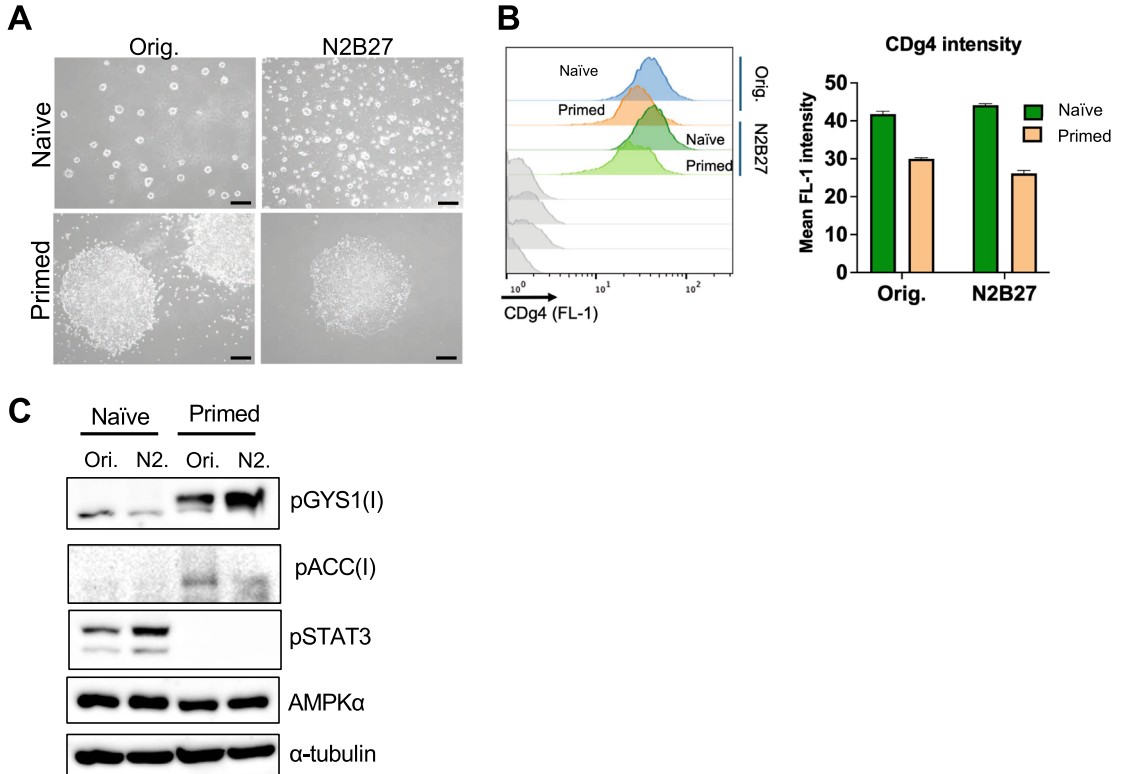

**Figure EV2. Higher glycogen amount and increased GYS1 activity in Naïve ESCs.**

(A) Phase contrast images of naïve (J1) and primed (PJ1) mESCs cultured under DMEM-based media (Org.) or chemically defined N2B27-based media (N2B27). Scale bar = 100 μm. (B) Flow cytometric analysis CDg4 with either Org. or N2B27-cultured naïve (J1) and primed (PJ1) mESCs. Quantified mean intensity of CDg4 is shown in right-side graph. (C) Immunoblotting analysis of naïve (J1) and primed (PJ1) mESCs cultured with either Org. or N2B27-mediated defined media (N2.). Indicative protein expression of pGYS1, pACC, pSTAT3 and AMPKα is shown, and α-tubulin was used for loading control. Active or inactive phosphorylation of each protein is marked as [A] or [I], respectively.

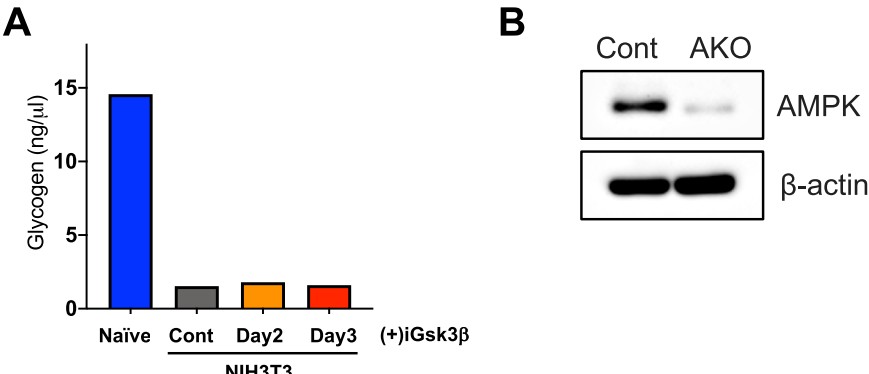

**A**

Glycogen (ng/µl)

Naïve | Cont | Day2 | Day3 | (+)iGsk3β

NIH3T3

**B**

Cont    AKO

AMPK

β-actin

Figure EV3.   High glycogen amount in Naïve ESCs, Establishment of AMPK KO cell line.

(**A**) Glycogen amount of naïve (J1) and NIH3T3 cells treated with iGSK3β for 3 days are quantified. (**B**) Immunoblotting analysis from Cont (J1) and Ampk KO J1 mESCs (AKO) with Ampk expression is shown, β-actin is used for internal loading control.

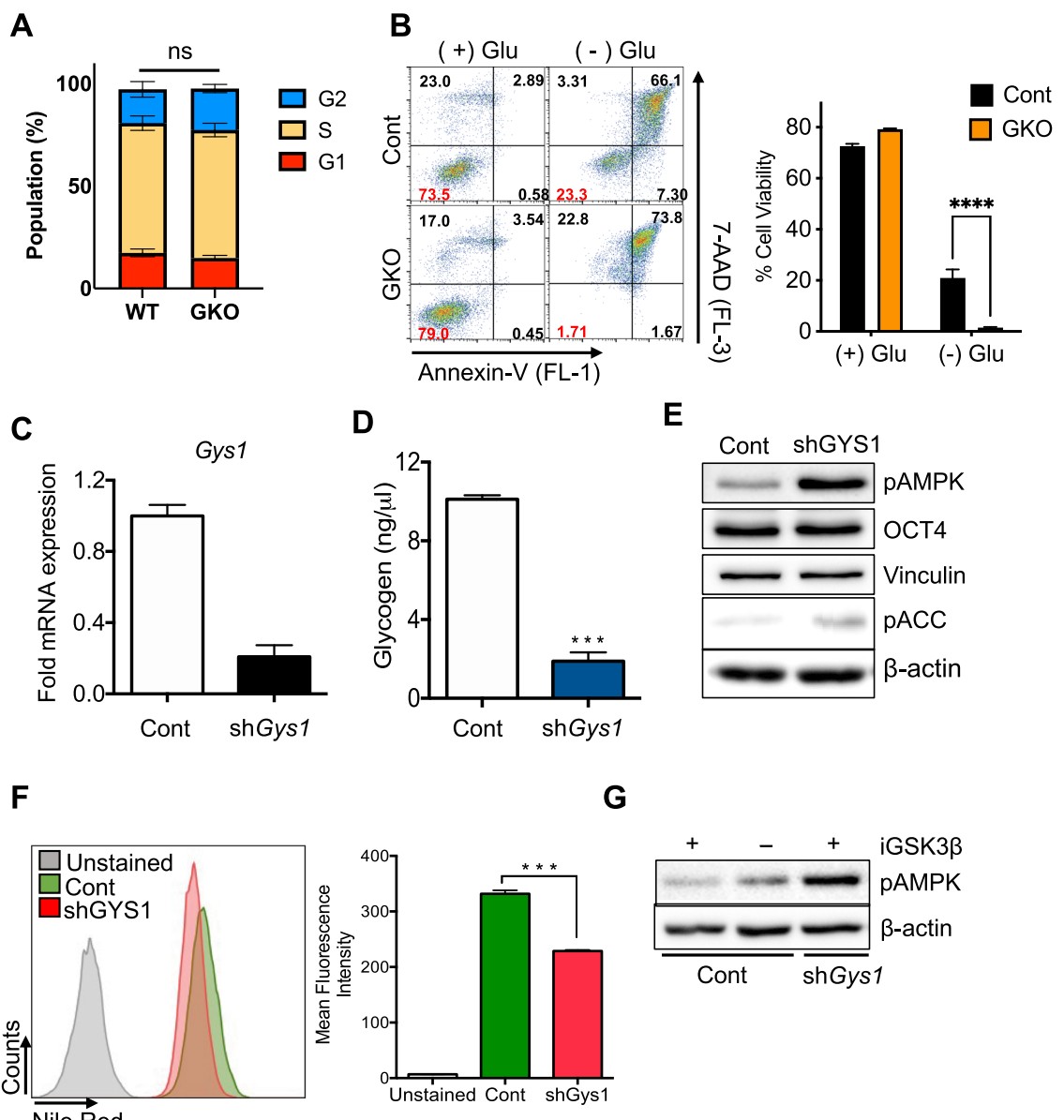

**Figure EV4. Establishment of Knockout (GKO) and Knockdown (shGys1) of Gys1.**

(A) Cell cycle profile analysis with WT (J1) and Gys1 KO (GKO) mESCs after treatment of Propidium iodide (PI)-staining is quantified. Based on DNA content estimation, each cell cycle stage (G1, S and G2) is analyzed. two-way ANOVA, multiple comparisons, ns indicates statistically not significant ($N = 3$, $n = 3$). (B) Flow cytometry for Annexin-V and 7-AAD staining of Cont and GKO with [+Glu] or without [-Glu] Glucose (left panel), quantification of live cell population (rignt panel). 2-way ANOVA, multiple comparisons, (****$P < 0.0001$, $N = 3$, $n = 3$). (C) Fold mRNA expression of Gys1 in before [Cont] and after transient knock down of Gys1 [shGys1]. (D) Intracellular glycogen level before [Cont] and after transient knock down of Gys1 [shGys1]. Multiple t-tests (***$P = 0.0003$, $N = 3$, $n = 3$). (E) Immunoblotting analysis for indicative proteins (pAmpkα, Oct4, Vinculin, pAcc, β-actin was used for loading control) before [Cont] and after transient knock down of Gys1 [shGys1]. (F) Flow Cytometry of Nile Red staining before [Cont] and after transient knock down of Gys1 [shGys1] (left panel), quantification of Mean Fluorescence Intensity (right panel). Multiple t-tests (***$P = 0.0002$, $N = 3$, $n = 3$). (G) Immunoblotting analysis for pAmpkα (β-actin was used for loading control) before [Cont] and after transient knock down of Gys1 [shGys1]. See also Fig. 3B.

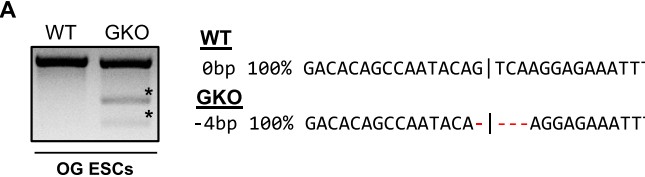

**A**

WT   GKO

**WT**
 0bp  100%  GACACAGCCAATACAG|TCAAGGAGAAATTT
**GKO**
-4bp  100%  GACACAGCCAATACA-|---AGGAGAAATTT

OG ESCs

**Figure EV5.  Establishment of *Gys1*KO (GKO) cell line.**

(**A**) T7E1 assay for WT and GKO in OG2$^{+/-}$GOF6$^{+/-}$ cell line (left panel), sequence information of targeted *Gys1* from WT and GKO in OG2$^{+/-}$GOF6$^{+/-}$ cell line (right panel).

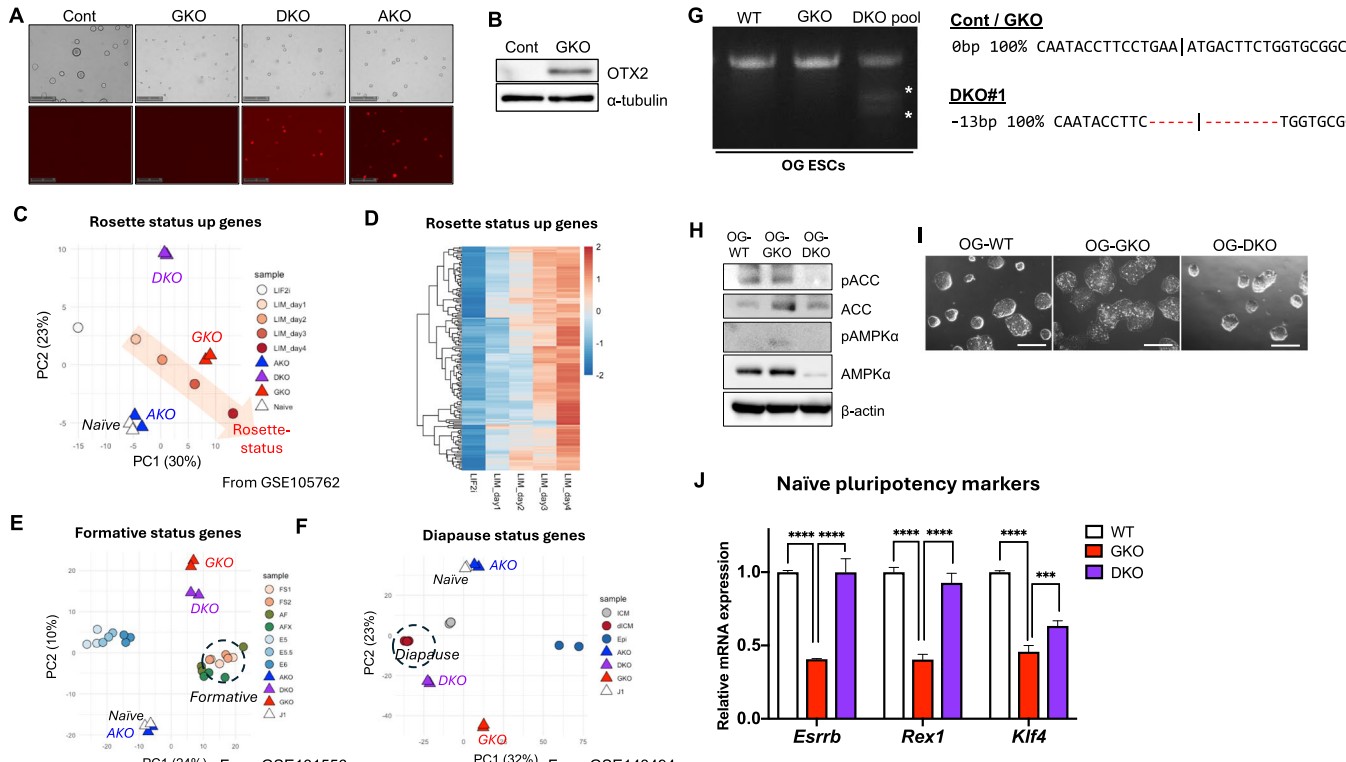

**Figure EV6. Establishment of *Gys1* KO (GKO) and *Prkaa1* KO (AKO).**

(A) Brightfield and RFP images for Cont, GKO, DKO and AKO (scale bar = 500 μm). (B) Immunoblotting analysis for indicative proteins (OTX2 in Cont and GKO, α-tubulin was used as a loading control). (C) PCA of RNA-seq data from this study (triangle shape; White: Naïve (J1), Blue: AKO, Purple: DKO and Red: GKO) and previously published studies (circle shape; Red: LIM medium, Blue: LFAI medium). Samples cultured in the medium containing LIF, PD325901, and the Wnt inhibitor IWP2 were labeled as LIM, with the duration of culture indicated numerically. Samples cultured in the medium containing LIF, FGF2, Activin A, and IWP2 were labeled as LFAI, similarly annotated with the duration of culture. Red arrows indicate changes observed in LIM medium under LIF+2i (PD325901, CHIR99021) conditions, while blue arrows indicate changes observed in LFAI medium under LIF+2i conditions. (D) Gene set variation analysis (GSVA) of Naive and primed pluripotency signature with LIF2i, LIM day1- day 4 samples from Fig. EV6C is shown. (E) PCA of RNA-seq results from this study; naïve (J1), AKO, DKO, and GKO mESCs and samples from GSE131556 (FS1, FS2, AF, AFX, E5, E5.5, E6) are shown. To induce formative status, Activin A (A), Fgf2 (F) or iWnt (X) is treated in the study GSE131556. Representative formative status samples are marked with dotted circle in the plot. (F) PCA from RNA-seq of mESCs in this study (naïve (J1), AKO, GKO and DKO) with diapause-like mESCs from GSE143492 (ICM, dICM, Epi). Diapause-like cells are marked with dotted circle in the plot. (G) T7E1 assay of *Prkaa1* gRNA target site for WT, GKO and DKO in OG-ESCs (left panel), sequence information of targeted *Prkaa1* from WT, GKO and DKO in OG-ESCs (right panel). Asterisks indicate the observed bands corresponding to expected size after T7E1 treatment. (H) Immunoblotting analysis with OG-WT, OG-GKO and OG-DKO mESCs with indicative proteins (pACC, total ACC, pAMPKα, total AMPK), β-actin is used as an internal loading control of the analysis. OG-ESCs: OG2$^{+/-}$GOF6$^{+/-}$ ESCs. (I) Representative brightfield images for Cont, GKO and DKO OG-ESCs (scale bar = 200 μm). (J) Fold mRNA expressions for *Esrrb*, *Rex1* and *Klf4* in WT, GKO and DKO OG-ESCs. Two-way ANOVA, multiple comparisons, (****$P < 0.0001$, ***$P = 0.0005$, $N = 3$, $n = 5$). OG-ESCs: OG2$^{+/-}$GOF6$^{+/-}$ ESCs.

