## [Peer Review File · EMBO Reports]

Ampk activation by glycogen expenditure primes the exit of naïve pluripotency

Seong-Min Kim, Eun-Ji Kwon, Ji-Young Oh, Han Sun Kim, Sunghyoun Park, Goo Jang, Jeong Tae Do, Keun-Tae Kim, and Hyuk-Jin Cha

Corresponding author(s): Keun-Tae Kim (kim.keuntae.7i@kyoto-u.ac.jp), Hyuk-Jin Cha (hjcha93@snu.ac.kr)

Review Timeline:

Transfer Date:	4th Mar 24
Editorial Decision:	4th Mar 24
Revision Received:	11th Oct 24
Editorial Decision:	6th Dec 24
Revision Received:	21st Dec 24
Accepted:	20th Jan 25

Editor: Deniz Senyilmaz Tiebe

Transaction Report: This manuscript was transferred to EMBO reports following peer review at The EMBO Journal.

Referee #1:

The manuscript by Seong-Min Kim, Eun-Ji Kwon and colleagues, investigate the role of Glycogen in naïve pluripotent stem cells. They found it is under the control of Glycogen synthase kinase 3. Downstream of glycogen they found AMPK, which, in turns regulates the levels of fatty acids. Most of the findings are not completely novel, and some have been already reported by the same authors. For example, the effect of GSK3 inhibitor on glycogen levels in primed and naïve cells (figure 3c) has been reported by the same group in 2022 (Kim et al, Metabolic Engineering, 2022). Figure 2B of Kim et al is identical to figure 3C of the current manuscript, which is rather confusing. The manuscript is often hard to read and there are several technical issues, listed below.

A major technical limitation of the study is that different media compositions have been used for naïve and primed cells (DMEM high glucose +15% FBS vs DMEM/F12 +20 KSR). It is obvious that differences in the composition will affect the metabolic profile of the cells. Simply the levels of Glucose could be enough to explain the differences in AMPK phosphorylation. All key experiments comparing naïve vs primed should be performed using the chemically defined medium N2B27, which is suitable for the culture of both cell types as described in dozen of publications in the last 15 years.

No statistics for 3C, 3H, 4G, 5B, 5G, 6G. The number and type of replicates is never defined. Were all experiments performed only once with technical replicates?

Figure 2E-G: the authors investigated only glucose as source of fatty acid synthesis. This is completely expected, but Acetyl-CoA could be obtained also from other sources, such as Glutamine, which is abundant in the medium. It is also expected that glucose deprivation will result in decreased Acetyl-CoA and a reduction of fatty acids, a simple consequence of energy deprivation. The authors should compare the contribution of different sources of Acetyl-Coa to draw conclusions.

Figure 2I: Acc inhibition causes cell death. Is this effect specific for naïve ESCs? Why is cell death a relevant readout, given that this was not observed in the case of glucose deprivation? The entire pathway described (GSK3 - glycogen - AMPK - FA) seems to regulate the transition from naïve to primed, why are the authors shifting the focus to cell death?

Figure 3B: the quality of the western blot is too low. Please show all 4 samples on the same membrane. pAMPK is very strong in Fig. 2B and barely detectable in fig. 3B, please explain.

Figure 3G-H: please show the markers for both WT and AKO, both in the presence and absence of GSK3i.

Figure 5D: there are two biological replicates for GKO cells. One is very close to naïve cells, the second one moves away from the trajectory followed by cells going from naïve to primed pluripotency. No meaning conclusion can be drawn from such data. Several studies identified marker genes and also signatures of naïve (E4.5), formative (E5.5) and

primed (E6.5) pluripotency states, for example Carbognin et al 2023 Nature Cell Biology, Kinoshita et al 2021 Cell Stem Cell, Kalkan et al 2019 Cell Stem Cell. The authors should show both RNA and protein levels of a large set of markers to define the phase of pluripotency of the GKO cells.

Figure 5G: Eomes is not expressed at biologically relevant levels in naïve cells, so any upregulation of such marker is not informative.

Figure 6I: AKO cells show increased fatty acids and glycogen, two molecular features that the authors linked to naïve pluripotency. However, AKO cells display reduced levels of KLF4 protein and increased OTX2 levels, indicating a reduction in naïve pluripotency.

Figure 6M: The results obtained with the dual reporter system are very problematic. WT cells should display a decrease of GFP+RFP- cells, in favour of GFP+RFP+. This is not the case even after 72h, actually the GFP+RFP- fraction increase. The whole dataset is therefore hard to interpret and trust. Furthermore, GFP+RFP+ cells should become RFP+GFP-, which is never the case. Simply put, if GKO cells are less naïve, they should differentiate faster towards the RFP+GFP- state.

Line 271-273: it is not clear why the authors describe results about Cdk8/19 inhibitors.

Line 311-315: please rephrase, it is very hard to understand.

Line 353-355: very convoluted, how can the levels of phosphorylation be attributed to levels of fatty acids in Fig. 4F, and to glucose/glycogen in fig. 2F-G?

Line 439-440: very hard to understand what 'loss of resilience to naïve conversion' means.

Referee #2:

In this manuscript, the authors reported a novel role of AMP-activated protein kinase (AMPK) in mouse embryonic stem cells (ESCs). Mouse ESCs possess the naïve pluripotency and transit to the ground state in the serum-free culture with 2 inhibitors (2i), MEK inhibitor and GSK3 inhibitor (Ying et al, 2008; Nichols and Smith, 2009). The authors found that the AMPK activity is inhibited by addition of 2i to the conventional culture with serum and LIF in parallel to MEK, ERK and GSK3. They revealed that the inhibition of the AMPK activity specifically occurs in the naïve state of pluripotent stem cells (PSCs) and that it mediates the fatty acid synthesis. The inhibition of AMPK is mediated by GSK3 inhibition via glycogen synthesis in a naïve PSC-specific manner. Elimination of Glycogen synthase 1 (Gys1) results in the activation of AMPK and reduction of the naïve-specific gene expressions, suggesting its role to control the exit from the naïve state.

It was reported that multiple metabolic transitions occur in the transition of PSCs from naïve to primed state. However, the finding shown in this manuscript looks novel and interesting. This is potentially worth to be published in EMBO J. However, the use of terminology surrounding the interpretation of these experimental results can be confusing. According to the original proposal by Nichols and Smith (2009), the naïve pluripotency is the pluripotent state found in the epiblast of pre-implantation embryos and mouse ESCs are at the naïve state in the conventional culture condition with serum and LIF. The serum-free culture with 2i captures ESCs at the ground state that

is defined as a sub-state of the naïve state and mimic the character of pre-implantation epiblast more precisely. In contrast, the authors use the term 'naïve state' for the ground state of naïve pluripotency and categorize the ESCs in the conventional culture into the metastable state. The term 'metastable' has been used for the description of unstable naïve state found in ESCs with particular genetic background (Hanna et al, CSC, 2009). Terms should be used according to their original definitions to avoid confusion.

In addition, there are several points required revision for publication.

1. Line 264: Although the authors stated that the cluster 3 exhibited a gradual decrease in phosphorylation, the list does not include GSK3 (Fig. 1D).
2. Fig. 2: Although the authors showed the comparisons between the naïve state (=ground state) and the primed state, the data of ESCs in LIF/Serum (conventional naïve state) should be included.
3. Fig. 5: The authors demonstrated that Gys1 KO ESCs are maintained at the distinct naïve state in 2i culture with serum and LIF. However, it is unclear whether they can grow at the ground state because the culture condition they applied allow the self-renewal in the conventional naïve state. The strict ground state culture without serum should be tested in this case.
4. Fig. 6: Double KO of Gys1 and AMPK reverse the phenotype of Gys1 KO ESCs only partially. What is the possible reason of this phenomenon?
5. Line 410: Correct gene symbol of the KO gene (Prkaa1) should be shown in the main text.

Referee #3:

The manuscript of Kim et al., presents evidence indicating that inhibition of AMPK activity by glycogen in ESCs results in elevated levels of fatty acids (FA) and stabilisation of the naïve state. Loss of glycogen derepresses AMPK and aids transition to the primed ESC state. The authors hypothesise that Glycogen/FA metabolism contributes to the naïve -to primed transition that occurs at embryo implantation. The authors build their case using publicly available phosphoproteomic data to uncover that AMPK activity is suppressed by 2iLIF culture - and that this is largely mediated by inhibition of GSK3. Adoption of the primed state is also associated with lower levels of FAs, and reduced levels of FA synthesis. Interestingly, deletion of AMPK allows the maintenance of the naïve state upon GSK3i withdrawal. However, as GSK3-mediated AMPK phosphorylation can in some circumstances be inhibitory the authors explore whether elevated Glycogen levels might suppress AMPK activity. Consistent with this idea, loss of Glycogen synthetase activity destabilises the naïve state, enforcing a more rapid transition to the primed like ESC phenotype, and can be prevented by depleting AMPK.

This manuscript provides interesting new insights into the role of metabolism in regulating pluripotency and how 2i conditions control the naïve state. Whilst the data is in the main clear and convincing, additional information is required to properly interpret the results, and some potential inconsistencies may need consideration.

Main points.

The results do not fully describe the conditions under which samples were collected, e.g. how long were the cells cultured following medium changes before sample collection? Might this variation

affect the degree of phosphorylation of AMPK etc in western blots? For example the levels of phosphorylation of AMPK in primed cells - compare the robust phosphorylation in Fig 2B, to apparently low levels in 2C (compared to b-actin control?) and 3B ? In addition, the negligible upregulation of pAMPK or pACC in Fig 6L, compared with more striking effects in Fig 4E, and even 6C? There are also inconsistencies in the total levels of AMPK between western blots, e.g. Fig 4E and Fig 6B?

Did control cells lines undergo similar subcloning regimes as the KO cells- and were they therefore subjected to equivalent rounds of clonal selection? How do the authors exclude the contribution of biases introduced through the process of selection and subcloning? Transgene based rescue experiments could eliminate these concerns and might also allow assessment of differentiation potential of (AMPK) KO clones.

Indeed, the identity of the replicate clones tested for each knock-out isn't clearly identified? How many were tested, and how consistent were they? In this regard what accounts for the apparent variation in the morphologies of the different GKO cells in experiments e.g. in Figure 5A/5F/6F, the GKO forms flattened colonies, but in Figure 6J, the GKO colonies have the similar size and compact morphology of the control cells, and again in Figure 5I, the OG-GKO colonies are smaller and more compact than the WT control colonies. What accounts for this variation? In fact the authors draw attention to the morphology in Fig 5A as a noteworthy measure of naïve phenotype.

How do the growth rates, or cell cycle profiles compare between the WT and KO cell lines? Could changes in the cell cycle (a lengthening G1?) or growth rates caused by metabolic shifts be responsible for changes in their tendency to differentiate?

LIF+Meki has been reported to be sufficient to maintain the naïve state (Dunn et al., Science doi: 10.1126/science.1248882) - The status of AMPK activity, Glycogen levels and FAs under these conditions would be interesting ?

More minor points

It would help to interpret the figures if the legends state exactly what the error bars in all figures represent, be it biological/independent experimental or technical replicates.

Are the fluorescent embryo images taken using a confocal microscope? It would be interesting to see the level of FAs in earlier and later embryos in direct side-by-side comparison.

Why is serum included in the naïve conditions? The standard N2B27 based 2iLIF medium does not require serum, and would be more directly comparable to a N2B27 based Activin/FGF regime for primed cells. The presence of serum could complicate comparisons between metabolism in different states of pluripotency.

Can the authors comment on the upregulation of Otx2 and apparent downregulation of KLF4 in the AKO cell lines in figure 6I?

In the results (line 299) the authors write "results indicate that the constant activation of AMPK, and the concurrent elevation of fatty acids are typical features of naïve mESCs" - I guess they mean "constant (in)activation" as it contradicts their findings and contrasts with the following sentence on line 304.

RE comments in the discussion and the Graphical Summary diagram it is worth considering that embryos of livestock species such as pig and cattle undergo the very first stages of gastrulation prior to embryo implantation.

Dear Keuntae,

Thank you for transferring your manuscript to EMBO Reports, which was previously reviewed at another journal.

Having read the manuscript and the referee reports, I would like to invite you to submit a revised manuscript to EMBO Reports as discussed before. In particular,

- Referee #1 notes a potential re-use of Figure 3C, which he/she states that it seems identical to Figure 2C of Kim et al, 2022a. Dual publication of a figure without clear citation is inadmissible. Instead, Figure 2C should be removed and Kim et al, 2022a should be cited.
 - Different medium compositions were used for naïve and primed cells, which renders the metabolic status incomparable. Key findings need to be repeated with the same culture medium (referee #1, paragraph starting as 'A major technical limitation...').
 - Other Acetyl-CoA sources than glucose need to be taken into consideration (referee #1, paragraph starting as 'Figure 2E-G:')
 - The stem cell states need to be named more carefully and other controls/markers need to be presented to support conclusions (referee #1, paragraph starting as 'Figure 5D:...'; referee #2, standfirst, points 2 and 3; referee #3, all main points)
 - The results of the dual reporter system are currently inconclusive (referee #1, paragraph starting as Figure 6M:').
 - Statistics and data presentation should be improved (all referees).
- Addressing the concerns of referee #3 regarding the mechanism of how basal feet are relevant to basal body alignment is not required for publication in EMBO Reports.

Should you be able to address the referee concerns in full, we would like to invite you to submit a revised manuscript. Please revise your manuscript with the understanding that the referee concerns (as in their reports) must be fully addressed and their suggestions taken on board. Please address all referee concerns in a complete point-by-point response. Acceptance of the manuscript will depend on a positive outcome of a second round of review. It is EMBO reports policy to allow a single round of major experimental revision only and acceptance or rejection of the manuscript will therefore depend on the completeness of your responses included in the next, final version of the manuscript.

We realize that it is difficult to revise to a specific deadline. In the interest of protecting the conceptual advance provided by the work, we recommend a revision within 3 months. Please discuss the revision progress ahead of this time with me if you require more time to complete the revisions, or if you have questions or comments regarding the revision (also by video chat).

1. A data availability section providing access to data deposited in public databases is missing (where applicable).
2. Your manuscript contains statistics and error bars based on $n=2$. Please use scatter plots in these cases.

You can submit the revision either as a Scientific Report or as a Research Article. For Scientific Reports, the revised manuscript can contain up to 5 main figures and 5 Expanded View figures, and it should not exceed 27000 characters. If the revision leads to a manuscript with more than 5 main figures it will be published as a Research Article. In this case the Results and Discussion section should be separate. If a Scientific Report is submitted, these sections have to be combined. This will help to shorten the manuscript text by eliminating some redundancy that is inevitable when discussing the same experiments twice. In either case, all materials and methods should be included in the main manuscript file.

3) We replaced Supplementary Information with Expanded View (EV) Figures and Tables that are collapsible/expandable online. A maximum of 5 EV Figures can be typeset. EV Figures should be cited as 'Figure EV1, Figure EV2' etc... in the text and their respective legends should be included in the main text after the legends of regular figures.

4) a .docx formatted letter INCLUDING the reviewers' reports and your detailed point-by-point responses to their comments. As part of the EMBO publication's Transparent Editorial Process, EMBO reports publishes online a Review Process File (RPF) to accompany accepted manuscripts. This File will be published in conjunction with your paper and will include the referee reports, your point-by-point response and all pertinent correspondence relating to the manuscript.

<https://www.embopress.org/page/journal/14693178/authorguide#transparentprocess>

5) a complete author checklist, which you can download from our author guidelines

<https://www.embopress.org/page/journal/14693178/authorguide>. Please insert information in the checklist that is also reflected in the manuscript. The completed author checklist will also be part of the RPF.

6) Please note that all corresponding authors are required to supply an ORCID ID for their name upon submission of a revised manuscript (<<https://orcid.org/>>). Please find instructions on how to link your ORCID ID to your account in our manuscript tracking system in our Author guidelines

<<https://www.embopress.org/page/journal/14693178/authorguide#authorshipguidelines>>

7) Before submitting your revision, primary datasets produced in this study need to be deposited in an appropriate public database (see <https://www.embopress.org/page/journal/14693178/authorguide#datadeposition>). Please remember to provide a reviewer password if the datasets are not yet public. The accession numbers and database should be listed in a formal "Data Availability" section placed after Materials & Method (see also

<https://www.embopress.org/page/journal/14693178/authorguide#datadeposition>). Please note that the Data Availability Section is restricted to new primary data that are part of this study. * Note - All links should resolve to a page where the data can be accessed. *

Additional information on source data and instruction on how to label the files are available:

<https://www.embopress.org/page/journal/14693178/authorguide#sourcedata>

9) Our journal encourages inclusion of *data citations in the reference list* to directly cite datasets that were re-used and obtained from public databases. Data citations in the article text are distinct from normal bibliographical citations and should directly link to the database records from which the data can be accessed. In the main text, data citations are formatted as follows: "Data ref: Smith et al, 2001" or "Data ref: NCBI Sequence Read Archive PRJNA342805, 2017". In the Reference list, data citations must be labeled with "[DATASET]". A data reference must provide the database name, accession number/identifiers and a resolvable link to the landing page from which the data can be accessed at the end of the reference. Further instructions are available at <http://www.embopress.org/page/journal/14693178/authorguide#referencesformat>

10) Regarding data quantification (see Figure Legends:

<https://www.embopress.org/page/journal/14693178/authorguide#figureformat>)

- the name of the statistical test used to generate error bars and P values,

- the number (n) of independent experiments (please specify technical or biological replicates) underlying each data point,

- the nature of the bars and error bars (s.d., s.e.m.),

- If the data are obtained from n Program fragment delivered error ``Can't locate object method "less" via package "than" (perhaps you forgot to load "than"?) at //ejpvfs23/sites23b/embor_www/letters/embor_decision_revise_and_review.txt line 56.' 2, use scatter blots showing the individual data points.

12) Please also note our reference format:

I look forward to seeing a revised version of your manuscript when it is ready. Please let me know if you have questions or comments regarding the revision.

Kind regards,

Deniz

Deniz Senyilmaz Tiebe, PhD
Editor
EMBO Reports

Kim & Kwon, et al., Response to Reviewers

Summary

We sincerely appreciate the reviewers' comments on our previous manuscript, which have greatly enhanced its quality. Below, our responses to the reviewers are highlighted in blue. Additionally, we have incorporated substantial new data along with corresponding explanations to provide a clear and concise summary of the updated information. We hope that the revised manuscript and our point-by-point responses to each comment will address the reviewers' concerns and clarify the key messages we intended to convey throughout this study.

In the revised manuscript and rebuttal, we have added significant new data with detail explanations, including;

1. Reproducibility of data using chemically defined media (i.e., N2B27-based media) cultured naïve and primed mESCs (Rebuttal Figs. 1 and 8B).
2. Mass spectrometry analysis of multiple metabolites labeled with ^{13}C -glucose and ^{13}C and ^{15}N -glutamine (Rebuttal Fig. 3).
3. Re-analysis of RNA-seq data for naïve, GKO, AKO, DKO and primed mESCs, with reduced intracellular variability (Rebuttal Fig. 5).
4. Detail explanation of utilizing Oct4-dual reporter lines for naïve and primed pluripotency (Rebuttal Fig. 7).
5. Re-performing immunoblot analyses to obtain clearer gel images (Rebuttal Figs. 4 and 8A).

As shown in this rebuttal, we made point-by-point response from each reviewer comment attached with a corresponding "Rebuttal Figure" in below. We strongly believe that the reviewers' comments have substantially improved our study. We hope the reviewers will concur that our work represents a meaningful discovery concerning the role of the glycogen-mediated *de novo* fatty acid synthesis pathway in maintaining naïve pluripotency and in the early stages of embryonic development.

General comments

Referee #1 notes a potential re-use of Figure 3C, which he/she states that it seems identical to Figure 2C of Kim et al, 2022a. Dual publication of a figure without clear citation is inadmissible. Instead, Figure 2C should be removed and Kim et al, 2022a should be cited.

- We apologize for the oversight in using duplicated data previously included in our earlier study. Following the reviewer's suggestion, we have removed Figure 3C from the original manuscript and have properly cited Kim et al., 2022a. We appreciate the reviewer's attention to this detail and have taken steps to ensure accurate referencing in our revised submission

Different medium compositions were used for naïve and primed cells, which renders the metabolic status incomparable. Key findings need to be repeated with the same culture medium (referee #1, paragraph starting as 'A major technical limitation...').

- As the reviewer suggested, we cultured both naïve and primed mESCs using N2B27-based serum-free medium and conducted crucial experiments, including 1) glycogen quantification and 2) analysis of the phosphorylation state of Ampk and its downstream targets (e.g., Acc, GSK3b). The results were consistent with those obtained using our original naïve medium, reinforcing that our observations reflect the inherent characteristics of naïve mESCs, independent of the medium used.
- Other Acetyl-CoA sources than glucose need to be taken into consideration (referee #1, paragraph starting as 'Figure 2E-G:')
 - As per the reviewer's comment, we first evaluated the overall amount of Acetyl-CoA in both naïve and primed PSCs through mass spectrometry analysis, using ¹³C-labeled glucose and ¹³C-labeled glutamine. Our findings indicate that the majority of Acetyl-CoA synthesis was derived from glucose, with minimal contribution from glutamine. Notably, glucose-derived Acetyl-CoA was significantly elevated in naïve PSCs compared to primed PSCs (Rebuttal Fig. 3).

- The stem cell states need to be named more carefully and other controls/markers need to be presented to support conclusions (referee #1, paragraph starting as 'Figure 5D:..'; referee #2, standfirst, points 2 and 3; referee #3, all main points)
 - Thanks for the reviewers' comments, we agree on the points that stem cell states need to be clarified and carefully renamed in light with specific markers of certain stage of pluripotency. In this regard, we either correct or provide additional supportive data to explain this as below.
 - As per the comment, we re-performed RNA-seq analysis with naïve, primed and GKO, AKO, DKO cells, with less intra-sample variations (Fig. 6I). Of note, it was consistent that transcriptomic profile of GKO is apart from that of naïve PSCs, indicating that it is located at intermediate state in between from naïve and primed states (Figs. 6I and S6C-F).
 - In order to clarify specific intermediate status of GKO, AKO and DKO cells, we extrapolated each cell transcriptome with multiple intermediate states, including formative (Nichols and Ying, 2006), diapause state (Huessein et al, 2020) and rosette state (Neagu et al, 2020). Interestingly, GKO seemed similar to rosette-pluripotency status (Figs. S6C and S6D) expressing higher OTX2 protein expression with KLF4 (Figs. 6H and S6B), which is typical molecular indication of rosette-pluripotent status (Neagu et al, 2020).
- The results of the dual reporter system are currently inconclusive (referee #1, paragraph starting as Figure 6M:').
 - In this study, we utilized the dual reporter system, where GFP and TdTomato (RFP) are regulated by the distal enhancer (DE) and proximal enhancer (PE) of *Pou5f1*, respectively. Since GFP marks naïve pluripotency, the transitions starting from GFP only, then GFP+RFP+, RFP only, and finally GFP-RFP- indicate the states of naïve, intermediate, primed, and differentiated, respectively (Rebuttal Figure 7A). To validate this, we recently developed a culture condition that promotes a gradual transition between naïve and primed pluripotency, reflected by these color changes. Under these conditions, clear gene

expression patterns corresponding to both naïve and primed pluripotency were seen, aligning with the stepwise color changes (Rebuttal Figure 7B).

- Therefore, in a naïve state, WT cells expressing GFP+RFP+ should gradually shift to display GFP only. However, the elevated proportion of GFP+RFP+ cells in GKO cells, even under naïve conditions, was significantly diminished. Notably, this effect was partially reversed in DKO cells (Fig. 6M). For further clarification, we included an additional figure (Rebuttal Figure 7C) showing the actual cell populations of GFP only (representing naïve) and GFP+RFP+ (representing intermediate) at specific time points.

- Statistics and data presentation should be improved (all referees).
 - As per all the reviewer comments, we improved all the statistical information in this updated version of manuscript.

Specific Comments

Referee #1:

The manuscript by Seong-Min Kim, Eun-Ji Kwon and colleagues, investigate the role of Glycogen in naïve pluripotent stem cells. They found it is under the control of Glycogen synthase kinase 3. Downstream of glycogen they found AMPK, which, in turns regulates the levels of fatty acids. Most of the findings are not completely novel, and some have been already reported by the same authors. For example, the effect of GSK3 inhibitor on glycogen levels in primed and naïve cells (figure 3c) has been reported by the same group in 2022 (Kim et al, Metabolic Engineering, 2022). Figure 2B of Kim et al is identical to figure 3C of the current manuscript, which is rather confusing. The manuscript is often hard to read and there are several technical issues, listed below.

- We thank the reviewer for thoughtful comments on our manuscript and sincerely apologize the oversight of duplicating the data between this manuscript and that of Kim et al., 2022a. Based on the suggestion of the reviewer, we removed Figure 3C from our document and have appropriately referenced the paper (Kim *et al*, 2022b).
- This manuscript builds upon our previous study, where we initially reported elevated intracellular glycogen levels in naïve ESCs (Kim *et al.*, 2022b). The present work aims to further explore the functional role of intracellular glycogen in naïve ESCs. We acknowledge that while some findings may not be entirely novel, this study provides a more comprehensive understanding of the role and downstream pathways of glycogen, including the involvement of AMPK in regulating fatty acid levels. We hope these revisions and clarifications address the reviewer's concerns, and we remain committed to ensuring the clarity and accuracy of our manuscript.

A major technical limitation of the study is that different media compositions have been used for naïve and primed cells (DMEM high glucose +15% FBS vs DMEM/F12 +20 KSR). It is obvious that differences in the composition will affect the metabolic profile of the cells. Simply the levels of Glucose could be enough to explain the differences in AMPK phosphorylation. All key experiments comparing naïve vs primed should be performed using the chemically defined

medium N2B27, which is suitable for the culture of both cell types as described in dozen of publications in the last 15 years.

- We appreciate the reviewer's valuable comments. To address the concern regarding the variation of glucose amounts in naïve and primed culture media, we used N2B27-based chemically defined, serum-free media to maintain naïve and primed mESCs, supplemented with LIF+2i for naïve mESCs and bFGF/Activin A for primed mESCs. The colony morphologies of naïve and primed ESCs cultured in N2B27-based medium (N2B27) remained consistent with those observed in the original medium (Orig.) (Fig. S2A). The initial observation of higher glycogen levels, indicated by CDg4 staining, was also reproduced in the N2B27-based medium (Fig. S2B) and this was linked to a decrease in the inhibitory phosphorylation of glycogen synthase [pGYS1(I)] (Fig. S2C). Similarly, the higher AMPK activity observed in primed ESCs, as indicated by the inhibitory phosphorylation of ACC [pACC(I)], was reproduced in the N2B27-based medium. Additionally, the production of acetyl-CoA from glucose (Rebuttal Fig. 1D), but not from glutamine (Rebuttal Fig. 1G), was consistent in the N2B27-based medium. The mass spectrometry results have been fully detailed in Rebuttal figure 1, as provided below.

Figure for referees not shown.

No statistics for 3C, 3H, 4G, 5B, 5G, 6G. The number and type of replicates is never defined. Were all experiments performed only once with technical replicates?

- We apologize for the missing statistics. We updated the statistics accordingly and added detail description of statistics at corresponding figure legends and method section.

Figure for referees not shown.

Figure 2E-G: the authors investigated only glucose as source of fatty acid synthesis. This is completely expected, but Acetyl-CoA could be obtained also from other sources, such as Glutamine, which is abundant in the medium. It is also expected that glucose deprivation will result in decreased Acetyl-CoA and a reduction of fatty acids, a simple consequence of energy deprivation. The authors should compare the contribution of different sources of Acetyl-CoA to draw conclusions.

- We appreciate the reviewer's comment. In order to precisely determine the level of acetyl-CoA from either glucose or glutamine, ^{13}C glucose or $^{13}\text{C},^{15}\text{N}$ glutamine was substituted for

normal glucose to naïve and primed ESCs for 24 hours (see Method section for detail). Upon ^{13}C glucose treatment, CoA was labeled with up to seven ^{13}C (majority 5 and 6 carbon) in both naïve and primed ESCs (Rebuttal Fig. 3A), which may be derived from ATP with similar number of ^{13}C (Rebuttal Fig. 3D). Given acetyl-group from glycolysis should be two ^{13}C , the final number of labeled carbons in Acetyl-CoA should be M+7, 8 and 9 (Rebuttal Fig. 3B). As predicted, acetyl-CoA mainly produced in naïve ESCs contained M+7, 8 and 9 (Rebuttal Fig. 3C). These results demonstrate that acetyl-CoA produced in naïve ESCs are mostly derived from glucose.

In similar approach, naïve and primed ESCs with ^{13}C , ^{15}N labeled glutamine, produced ATP and CoA with M+1, 2, 3, 4 and 5 (Rebuttal Figs. 3E and F). Unlike the case of ^{13}C glucose labeling, the acetyl CoA from ^{13}C , ^{15}N labeled glutamine only had M+2, 3, 4 and 5 (Rebuttal Fig. 3G), suggesting that no acetyl-group derived from glutamine contributes to producing Acetyl-CoA in both naïve and primed ESCs. These results were fully addressed in the result section accordingly.

Figure for referees not shown.

Figure 2I: Acc inhibition causes cell death. Is this effect specific for naïve ESCs? Why is cell death a relevant readout, given that this was not observed in the case of glucose deprivation? The entire pathway described (GSK3 - glycogen - AMPK - FA) seems to regulate the transition from naïve to primed, why are the authors shifting the focus to cell death?

- We appreciate the constructive comments from the reviewer. As briefly described in the introduction (page 3, lines 48-52), the knockout of key genes involved in fatty acid synthesis, such as *Fasn* and *Acc1* (Abu-Elheiga *et al*, 2005; Chirala *et al*, 2003), induces embryonic lethality. This is further highlighted by the cell death observed in naïve ESCs following ACC1 inhibitor treatment (Fig. 2J). While the requirement for fatty acid synthesis in ESC survival or pluripotency (Wang *et al*, 2017) is not fully understood, one study has shown that fatty acid oxidation can rescue ESCs from metabolic stress (Yan *et al*, 2021). The result shown in Figure 2J would be another evidence to highlight the high demand of fatty acid synthesis in naïve pluripotency.

Figure 3B: the quality of the western blot is too low. Please show all 4 samples on the same membrane. pAMPK is very strong in Fig. 2B and barely detectable in fig. 3B, please explain.

- We regret to admit that the multiple pAMPK antibodies we purchased from the same vendor and different provider (Cell Signaling Technologies Cat#2535 and ThermoFisher Cat#44-1150G) failed to produce as strong a signal as the older pAMPK antibody. Despite number of attempts, we could not resolve the technical difficulty due to possible variation from the different lot# of antibody. Thus, we acknowledge that the weak pAMPK band observed with the newly purchased antibody is the best result we could obtain. However, the distinct phosphorylation of ACC (at serine 79), a key downstream event of AMPK activation, in primed ESCs provides strong evidence of AMPK activation. In addition, we added new results to demonstrate the active GSK3 β [pGSK3 β (A)], corresponding the inhibitory GYS1 [pGYS1(I)] and phosphorylation dependent electrophoretic mobility shift (PDEMS) of GYS1 (Fig. 2B) marked by arrows.

Figure for referees not shown.

Figure 5D: there are two biological replicates for GKO cells. One is very close to naïve cells, the second one moves away from the trajectory followed by cells going from naïve to primed pluripotency. No meaning conclusion can be drawn from such data.

- We admitted the reviewer's kind comment and removed Figure 5D where the results were not addressable. However, PCA plot presented in Figure 5C and D, was clear enough to address the difference of GKO compared to naïve control. In order to address the following comment from the reviewer, we achieved the new transcriptome dataset with naïve, AKO, DKO and primed ESCs that may minimize the intra variability (RNA-seq data deposited GSE272593). Then, the transcriptome datasets were thoroughly analyzed to determine the effect of AKO on reversion of GKO characteristics. As shown in Figure 6G and H, alteration of transcriptome signature of GKO was partly restored in DKO.

Several studies identified marker genes and also signatures of naïve (E4.5), formative (E5.5) and primed (E6.5) pluripotency states, for example Carbognin et al 2023 Nature Cell Biology, Kinoshita et al 2021 Cell Stem Cell, Kalkan et al 2019 Cell Stem Cell. The authors should show both RNA and protein levels of a large set of markers to define the phase of pluripotency of the GKO cells.

- As per the reviewer's comment, there are multiple studies, characterizing the intermediate states between naïve and primed pluripotency based on transcriptome dataset, including formative (Kinoshita *et al*, 2021), rosette (Neagu *et al*, 2020), and diapause (Hussein *et al*, 2020) stage. We compared the transcriptome signature of the set of naïve, AKO, GKO, and DKO ESCs (Figs. 6G and 6H) to those of rosette (Fig. S6C and S6D), formative (Fig. S6E), and diapause (Fig. S6F) stage to present the similarity in PCA plots after normalization (Rebuttal Fig. 5A). Of interest, transcriptome signature of GKO cells were similar to that of rosette stage, which was partly reversed in DKO cells (Fig. S6C).
- As previously demonstrated that WNT activity serves as a major determinant between naïve pluripotency and rosette stage (Neagu *et al.*, 2020), Wnt activity was significantly reduced in GKO cells than control naïve ESCs (Rebuttal Fig. 5B). The downregulation of Wnt activity would be associated to the induction of Otx2, a key transcription factor, expressed in rosette stage (Neagu *et al.*, 2020) and GKO cells (Fig. 6G and 6H). We are currently investigating the biological consequence of reduced WNT activity in GKO and the mechanism. Thus, this preliminary result is provided to the reviewer only as a form of addendum figure.

Figure for referees not shown.

Figure 5G: *Eomes* is not expressed at biologically relevant levels in naïve cells, so any upregulation of such marker is not informative.

- As per the reviewer's comment, we substituted the mRNA expression of *Eomes* to *T* (*T Brachyury transcription factor*). The distinct induction of *Fgf5* and *T* after glucose deprivation in GKO cells, indicated the attenuation of naïve pluripotency and transition to primed state and differentiation on the basis of induction of *Fgf5* and *T* expression (Rebuttal Fig. 6). Text was slightly edited accordingly.

Figure for referees not shown.

Figure 6I: AKO cells show increased fatty acids and glycogen, two molecular features that the authors linked to naïve pluripotency. However, AKO cells display reduced levels of KLF4 protein and increased OTX2 levels, indicating a reduction in naïve pluripotency.

- We appreciate the reviewer's comment. OTX2 protein, upregulated in GKO cells were markedly reduced in DKO (Fig. 6H), which was corresponding to the mRNA levels (Fig. 6G). The protein level of OTX2 in AKO was distinctly lowered than that of GKO. However, we noticed that KLF4 protein levels were unexpectedly affected in AKO cells. One possible explanation of the decrease of KLF4 protein level in AKO would be the role of AMPK itself for KLF4 expression as previously described that KLF4 is a downstream target of AMPK (Sunaga *et al*, 2016). According to the transcriptome signature analysis (Figs. 5C, 6I and J)

as well as the dome-shaped colony morphology (Fig. 6F), it is clear that AKO cells maintained naïve pluripotency. We briefly described this in the result section.

Figure 6M: The results obtained with the dual reporter system are very problematic. WT cells should display a decrease of GFP+RFP- cells, in favour of GFP+RFP+. This is not the case even after 72h, actually the GFP+RFP- fraction increase. The whole dataset is therefore hard to interpret and trust. Furthermore, GFP+RFP+ cells should become RFP+GFP-, which is never the case. Simply put, if GKO cells are less naïve, they should differentiate faster towards the RFP+GFP- state.

- We appreciate the reviewer's feedback. As noted, GFP and TdTomato (RFP) are controlled by the distal enhancer (DE) and proximal enhancer (PE) of *Pou5f1*, respectively, as described by Choi et al., 2016 (Choi *et al*, 2016). GFP-only (GFP+RFP-) indicates naïve pluripotency. Therefore, as illustrated in Rebuttal Figure 7A, the transition from GFP-only, GFP+RFP+, RFP-only, to GFP-RFP- reflects a unidirectional progression from naïve, intermediate, and primed pluripotency to differentiation.
- To further investigate this, we recently developed a culture condition designed to induce gradual transitions between naïve and primed pluripotency, corresponding to the color changes. Under these conditions, we observed a clear parallel between the gradual color shift and the gene expression patterns associated with naïve and primed states (Rebuttal Figure 7B).
- In line with this, under naïve conditions, WT cells expressing GFP+RFP+ should gradually shift to GFP+ only. While a higher proportion of GFP+RFP+ cells remained in GKO cells even under naïve conditions, this phenomenon was significantly reduced. Notably, this effect was partly reversed in DKO cells (Figure 6M). To clarify this observation, we have provided an additional figure (Rebuttal Figure 7C) showing the actual cell populations of GFP-only (indicating naïve pluripotency) and GFP+RFP+ (representing the intermediate state) at the indicated time points.

Figure for referees not shown.

Line 271-273: it is not clear why the authors describe results about Cdk8/19 inhibitors.

- To achieve naïve pluripotency, other than supplement of 2i (MEK1/GSK3 β inhibitors), Cdk8/19 inhibitors treatment induces naïve pluripotency more effectively (Lynch *et al*, 2020). As the time dependent phosphoproteome data after Cdk8/19 inhibitor treatment is

available (Martinez-Val *et al*, 2021), it would be another dataset for highlighting the AMPK inhibition in naïve ESCs. The text was edited accordingly.

Line 311-315: please rephrase, it is very hard to understand.

- The text was modified as below. *“It is noteworthy that while GSK3 β directly inhibits AMPK in somatic cell models (Suzuki *et al*, 2013), inhibition of GSK3 β led to the inhibition of AMPK in naïve ESCs (Fig. 3B) The unexpected AMPK activity, suggests that other, yet unidentified, factors may contribute to the observed inhibition of AMPK by iGsk3 β specifically in naïve ESCs.”*

Line 353-355: very convoluted, how can the levels of phosphorylation be attributed to levels of fatty acids in Fig. 4F, and to glucose/glycogen in fig. 2F-G?

- We rephrased the according sentence as below. *“As ACC is a key regulator of fatty acid levels, promoting fatty acid synthesis and inhibiting fatty acid oxidation, its inhibition—indicated by ACC phosphorylation in GKO (Fig. 4E)—would lead to a decrease in intracellular fatty acids, as shown by BODIPY staining (Fig. 4F).”*

Line 439-440: very hard to understand what 'loss of resilience to naïve conversion' means.

- We had intended to address that GKO cells were like to be less susceptible to naïve conversion unlike WT. As per the reviewer’s kind comment, we edited the text as below. *“Flow cytometry analysis demonstrated that glycogen is essential for the efficient conversion to naïve pluripotency (Fig. 5J).”*

Referee #2:

In this manuscript, the authors reported a novel role of AMP-activated protein kinase (AMPK) in mouse embryonic stem cells (ESCs). Mouse ESCs possess the naïve pluripotency and transit to the ground state in the serum-free culture with 2 inhibitors (2i), MEK inhibitor and GSK3 inhibitor (Ying et al, 2008; Nichols and Smith, 2009). The authors found that the AMPK activity is inhibited by addition of 2i to the conventional culture with serum and LIF in parallel to MEK, ERK and GSK3. They revealed that the inhibition of the AMPK activity specifically occurs in the naïve state of pluripotent stem cells (PSCs) and that mediates the fatty acid synthesis. The inhibition of AMPK is mediated by GSK3 inhibition via glycogen synthesis in naïve PSC-specific manner. Elimination of Glycogen synthase 1 (Gys1) results in the activation of AMPK and reduction of the naïve-specific gene expressions, suggesting its role to control the exit from the naïve state.

It was reported that multiple metabolic transitions occur in the transition of PSCs from naïve to primed state. However, the finding shown in this manuscript looks novel and interesting. This is potentially worth to be published in EMBO J.

- We thank the reviewer for thoughtful comments on our manuscript, below we added our response to each point.

However, the use of terminology surrounding the interpretation of these experimental results can be confusing. According to the original proposal by Nichols and Smith (2009), the naïve pluripotency is the pluripotent state found in the epiblast of pre-implantation embryos and mouse ESCs are at the naïve state in the conventional culture condition with serum and LIF. The serum-free culture with 2i captures ESCs at the ground state that is defined as a sub-state of the naïve state and mimics the character of pre-implantation epiblast more precisely. In contrast, the authors use the term 'naïve state' for the ground state of naïve pluripotency and categorize the ESCs in the conventional culture into the metastable state. The term 'metastable' has been used for the description of unstable naïve state found in ESCs with particular genetic background (Hanna et al, CSC, 2009). Terms should be used according to their original definitions to avoid confusion. In addition, there are several points required revision for publication.

- We appreciate the reviewer's constructive comment. As the reviewer noted, the ground state is often characterized as a more uncommitted state, reflecting the features of the pre-implantation embryo. While several studies have sought to distinguish between the naïve and ground states, many others have not strictly differentiated between these terms when referring to naïve pluripotency maintained with 2i/LIF (Baker & Pera, 2018; Cornacchia *et al*, 2019; Gatchalian *et al*, 2018). To simplify our terminology, we defined the pluripotency states as 'naïve' and 'primed.'
- The term 'metastable,' used to describe the LIF-only condition where both GFP and RFP are expressed, acknowledges the study that introduced the 'dual reporter system,' which characterizes cell states based on GFP and RFP expression (Choi *et al.*, 2016). The 'metastable' state of GFP+/RFP+ cells was further examined through their gradual transition to either a naïve or primed state. As demonstrated in Rebuttal Figure 4, typical genes associated with naïve and primed pluripotency were upregulated in a time-dependent manner, correlating with the GFP/RFP fluorescence.
- We kindly ask for your understanding that these results, which are part of a separate research project currently under review at another journal, are provided here as an addendum.

Figure for referees not shown.

1. Line 264: Although the authors stated that the cluster 3 exhibited a gradual decrease in phosphorylation, the list does not include GSK3 (Fig. 1D).

- Thanks for the reviewer's comment. The original phospho-proteome dataset used for producing the results in Figure 1D (Martinez-Val *et al.*, 2021) lacks the downstream targets of GSK3 β , in which we believe the absence of GSK3 β in the list. For the clarification of timely inhibition of GSK3 β , two known substrates of GSK3 β were presented in Figure S1B.

2. Fig. 2: Although the authors showed the comparisons between the naïve state (=ground state) and the primed state, the data of ESCs in LIF/Serum (conventional naïve state) should be included.

- As per the reviewer comment, we examined the key signaling events by immunoblotting analysis. Under LIF/Serum [shown as (-)], GYS1 inhibitory phosphorylation [pGYS1(I)], a downstream event of GSK3 β [corresponding to phosphorylation dependent electrophoretic mobility shift (PDEMS) of GYS1], was higher than under LIF/2i condition. The complete inhibition of ERK1/2 and GSK3 β under LIF/2i compared to the primed state, was manifested (Fig. 2B; Rebuttal fig. 4).

Figure for referees not shown.

3. Fig. 5: The authors demonstrated that Gys1 KO ESCs are maintained at the distinct naïve state in 2i culture with serum and LIF. However, it is unclear whether they can grow at the ground state because the culture condition they applied allow the self-renewal in the conventional naïve state. The strict ground state culture without serum should be tested in this case.

- We appreciate the reviewer's critical comment. As highlighted by the reviewer at the beginning of the overall feedback, and in line with the original proposal from Nichols and

Smith (2009), ground state culture represents a very specific window of pre-implantation development whilst naïve pluripotency encompasses a somewhat broader concept within the pre-implantation stage of embryos.

- In response to the reviewer's point, we observed that naïve mESCs cultured in both Serum/2i+LIF (Orig.) media and ground state media (N2B27) containing serum replacement, exhibited only marginal differences in colonial morphologies (Fig. S2A), the level of glycogen (Fig. S2B), key signaling events (Fig. S2C), and Acetyl-CoA derived from glucose (Rebuttal fig. 8A).
- However, unlike WT naïve ESCs, Gys1 KO ESCs lost the naïve characteristics rapidly under the ground state culture conditions, determined by colonial morphology and gene expression (Rebuttal Figure 8B). This suggests that additional signaling modulators, such as Wnt inhibitors (Neagu et al, 2020), are required to sustain the Gys1 KO ESCs, indicating that these cells are more inclined to shift toward a later stage of pluripotency compared to the naïve pre-implantation stage epiblast.

Figure for referees not shown.

4. Fig. 6: Double KO of Gys1 and AMPK reverse the phenotype of Gys1 KO ESCs only partially. What is the possible reason of this phenomenon?

- Thank you for the reviewer's insightful comment, which we had not previously considered in depth. It is well-known that the carbohydrate-binding module (CBM), through which glycogen interacts with the β 1 subunit of AMPK, is also found in various other eukaryotic proteins, such as EPM2A (Gentry *et al*, 2013), STBD1 (Zhu *et al*, 2014). Therefore, the functions of proteins regulated by glycogen, aside from AMPK, cannot be fully addressed through the knockout of AMPK alone. In addition, glycogen directly regulates protein phosphatase 1 (PP1) activity via interaction with protein targeting to glycogen (PTG) (Semrau *et al*, 2022; Wu *et al*, 1998). This suggests that the loss of glycogen may induce unexpected effects that regulate various protein activities and signaling pathways, which may account for the incomplete recovery observed in the DKO cells from GKO mESCs.

5. Line 410: Correct gene symbol of the KO gene (*Prkaa1*) should be shown in the main text.

- As per the reviewer's comment, we added gene symbol of AMPK α (i.e., *Prkaa1*) over the main text.

Referee #3:

The manuscript of Kim et al., presents evidence indicating that inhibition of AMPK activity by glycogen in ESCs results in elevated levels of fatty acids (FA) and stabilisation of the naïve state. Loss of glycogen derepresses AMPK and aids transition to the primed ESC state. The authors hypothesise that Glycogen/FA metabolism contributes to the naïve -to primed transition that occurs at embryo implantation. The authors build their case using publicly available phosphoproteomic data to uncover that AMPK activity is suppressed by 2iLIF culture - and that this is largely mediated by inhibition of GSK3. Adoption of the primed state is also associated with lower levels of FAs, and reduced levels of FA synthesis. Interestingly, deletion of AMPK allows the maintenance of the naïve state upon GSK3i withdrawal. However, as GSK3-mediated AMPK phosphorylation can in some circumstances be inhibitory the authors explore whether elevated Glycogen levels might suppress AMPK activity. Consistent with this idea, loss of Glycogen synthetase activity destabilises the naïve state, enforcing a more rapid transition to the primed like ESC phenotype, and can be prevented by depleting AMPK. This manuscript provides interesting new insights into the role of metabolism in regulating pluripotency and how 2i conditions control the naïve state. Whilst the data is in the main clear and convincing, additional information is required to properly interpret the results, and some potential inconsistencies may need consideration.

Main points

The results do not fully describe the conditions under which samples were collected, e.g. how long were the cells cultured following medium changes before sample collection? Might this variation affect the degree of phosphorylation of AMPK etc in western blots? For example the levels of phosphorylation of AMPK in primed cells - compare the robust phosphorylation in Fig 2B, to apparently low levels in 2C (compared to b-actin control?) and 3B? In addition, the negligible upregulation of pAMPK or pACC in Fig 6L, compared with more striking effects in Fig 4E, and even 6C? There are also inconsistencies in the total levels of AMPK between western blots, e.g. Fig 4E and Fig 6B?

- We regret to admit that the multiple pAMPK antibodies we purchased from the same vendor and other sources failed to reproduce the strong signal observed with the older pAMPK antibody. Therefore, the less evident pAMPK levels in primed ESCs, particularly in previous Figures 3B, 6L and the newly added Figure 2B, represent the best results we could obtain. However, the distinct phosphorylation of ACC at serine79, a key downstream marker of AMPK activation, consistently observed in primed ESCs, provides strong evidence of significant AMPK activation in these cells.
- Regarding AMPK protein stability, we also observed occasional variations in AMPK protein levels despite comparable mRNA levels between naïve and primed ESCs (Rebuttal Figure 9A). As shown in Figure 2C, reduced AMPK protein levels in primed ESCs coincided with strong AMPK activity. Through literature review, we recognized that AMPK protein is subject to degradation (Pineda *et al*, 2015; Qi *et al*, 2008), as high energy input induces AMPK degradation via negative feedback mechanisms (Jiang *et al*, 2021). Thus, the most plausible explanation is that AMPK degradation occurs more frequently in primed ESCs, where high AMPK activity is observed. It is also important to note that the medium for both naïve and primed ESCs must be replaced daily due to their high glucose demand.

Figure for referees not shown.

Did control cells lines undergo similar subcloning regimes as the KO cells- and were they therefore subjected to equivalent rounds of clonal selection? How do the authors exclude the contribution of biases introduced through the process of selection and subcloning? Transgene based rescue experiments could eliminate these concerns and might also allow assessment of differentiation potential of (AMPK) KO clones.

- In order to exclude the bias of clonal variation, we used two independent sets of naïve, GKO, AKO, DKO and primed ESCs (J1 and OG2 ESCs). Results presented until the Figure 5, were produced from the set of J1 mESCs and results in Figure 5H, 6K-M were derived from the set of OG2 mESCs. The different set information was added in the figure legend.

Indeed, the identity of the replicate clones tested for each knock-out isn't clearly identified? How many were tested, and how consistent were they? In this regard what accounts for the apparent variation in the morphologies of the different GKO cells in experiments e.g. in Figure 5A/5F/6F, the GKO forms flattened colonies, but in Figure 6J, the GKO colonies have the similar size and compact morphology of the control cells, and again in Figure 5I, the OG-GKO colonies are smaller and more compact than the WT control colonies. What accounts for this variation?

- Thank you for raising this point. For the establishment of the KO lines, we performed clonal selection within that pool after introducing CRISPR/Cas9, and then selected more than 20 single clones for each target. The functional KO of selected clones was validated by the level of glycogen and CDg4 staining, a fluorescence probe for glycogen (Kim *et al.*, 2022b).
- A recent study demonstrates that the naïve characteristics and differentiation potential of naïve mESCs can vary depending on their genetic backgrounds (Ortmann *et al.*, 2020). Given that the J1 and OG2 lines have distinct genetic backgrounds, their morphological traits or differentiation potential may indeed differ. However, as long as the appropriate control experiments are properly conducted, the main focus of this study remains to highlight that *Gys1* KO mESCs are more prone to differentiation and exit from the naïve state compared to wild-type naïve mESCs.

In fact, the authors draw attention to the morphology in Fig 5A as a noteworthy measure of naïve phenotype. How do the growth rates, or cell cycle profiles compare between the WT and KO cell lines? Could changes in the cell cycle (a lengthening G1?) or growth rates caused by metabolic shifts be responsible for changes in their tendency to differentiate?

- We appreciate reviewer's interest in the cell characterization. As the reviewer already recognizes, cell cycle control in naïve ESCs is distinct from that of normal somatic cells (White & Dalton, 2005). The constant expression cyclin A and E, following CDK2 activation (Stead *et al*, 2002) promotes cell cycle lacking G1 and G2 checkpoint. Thus, we presumed that depletion of glycogen would only marginally affect the cell cycle profile. As predicted, no clear cell cycle alteration was observed (Fig. S4A; Rebuttal Fig. 10).

Figure for referees not shown.

LIF + Mek1 has been reported to be sufficient to maintain the naïve state (Dunn et al., Science doi: 10.1126/science.1248882) - The status of AMPK activity, Glycogen levels and FAs under these conditions would be interesting?

- As similar as the reviewer's comment, we had examined the independent effect of iMEK1 and iGSK3 β on naïve pluripotency. After withdrawal of either iMEK1 or iGSK3 β in WT naïve ESCs, a typical dome-shape morphology of naïve ESCs was clearly lost. Please disregard the result with KD naïve ESCs and focus on WT results, which was a part of the previous independent study (Kim *et al*, 2022c). The level of AMPK activity and intracellular glycogen

after iGSK3 β depletion (i.e., LIF+Meki) was already demonstrated in the previous study (Kim *et al*, 2022a) (Addendum Figure. 11A).

- In addition, we further analyzed the transcriptomic changes caused by the depletion of iMEK1 or iGSK3 β . As shown in Rebuttal Figure 11B, genes that were upregulated or downregulated due to the absence of iMEK1 (iGSK3 β only) or iGSK3 β (iMEK1 only) were compared to the control condition where both iMEK1 and iGSK3 β were present. Interestingly, gene sets highly associated with mesoderm differentiation (e.g., mesoderm development and muscle tissue development) were affected by the loss of iMEK1, while ectoderm/endoderm differentiation (e.g., renal, kidney, hindbrain, and axon guidance) was impacted by the loss of iGSK3 β . This suggests that proper differentiation into the three germ layers requires the simultaneous inhibition of both MEK1 and GSK3 β . Additionally, gene sets related to metabolism, such as amino acid metabolism, were notably downregulated upon GSK3 β depletion compared to the control, implying that GSK3 β suppression plays a critical role in the metabolic control of naïve ESCs.
- Although these preliminary findings are not directly relevant to the main focus of this manuscript, we provide them as an addendum for the reviewer's consideration.

Figure for referees not shown.

More minor points

It would help to interpret the figures if the legends state exactly what the error bars in all figures represent, be it biological/independent experimental or technical replicates.

- As per the reviewer's comment, we added detail statistical information of each figure at corresponding figure legends. Particular figures with revised statistical analysis are shown below (Rebuttal Figure 12).

Figure for referees not shown.

Are the fluorescent embryo images taken using a confocal microscope? It would be interesting to see the level of FAs in earlier and later embryos in direct side-by-side comparison.

- All fluorescent embryo images were captured using epi-fluorescent microscopy. Regarding the fatty acid levels across different stages of mouse embryos, we indeed attempted to culture mouse embryos up to the post-implantation stage (i.e., beyond E4.5). However, it

proved technically challenging to capture certain stages, maintain them in culture, and obtain a sufficient number of embryos for fatty acid quantification by mass spectrometry. Nevertheless, I agree that quantifying fatty acid levels in post-implantation embryos would be a valuable pursuit for future studies.

Why is serum included in the naïve conditions? The standard N2B27 based 2iLIF medium does not require serum and would be more directly comparable to a N2B27 based Activin/FGF regime for primed cells. The presence of serum could complicate comparisons between metabolism in different states of pluripotency.

- Corresponding to similar comments from the reviewer #1, we performed key experiments using N2B27-based, chemically defined medium (N2) to culture both naïve and primed mESCs and found the consistent results from those using original (Ori.) medium (Rebuttal Figure 1). We also described corresponding results in main text of the updated version of manuscript.

Figure for referees not shown.

Can the authors comment on the upregulation of Otx2 and apparent downregulation of KLF4 in the AKO cell lines in figure 6I?

- We appreciate the reviewer's comment. OTX2 protein, highly upregulated in GKO cells were markedly reduced in DKO (Fig. 6H), which was corresponding to the mRNA levels (Fig. 6G). The protein level of OTX2 in AKO was distinctly lowered than that of GKO. However, we noticed that KLF4 protein levels were unexpectedly affected in AKO cells. One possible explanation of the decrease of KLF4 protein level in AKO would be the role of AMPK itself for KLF4 expression as previously described that KLF4 is a downstream target of AMPK (Sunaga *et al.*, 2016). According to the transcriptome data (Figs. 6I and 6J) as well as the dome-shaped colony morphology (Fig. 6F), it is clear that AKO cells maintained naïve pluripotency. We briefly described this in the result section.

In the results (line 299) the authors write "results indicate that the constant activation of AMPK, and the concurrent elevation of fatty acids are typical features of naïve mESCs" - I guess they mean " constant (in)activation" as it contradicts their findings and contrasts with the following sentence on line 304.

- We thank the reviewer for catching this and updated this in the updated version of manuscript.

RE comments in the discussion and the Graphical Summary diagram it is worth considering that embryos of livestock species such as pig and cattle undergo the very first stages of gastrulation prior to embryo implantation.

- We appreciate the reviewer's kind suggestion. In the discussion, we briefly addressed the potential link between fatty acid levels and the preimplantation period in a few livestock species, drawing from both literature and results from mouse ESC models. While it is well-documented that reducing fatty acids by promoting fatty acid oxidation through L-Carnitine treatment in bovine and porcine embryos prior to artificial embryo transfer improves pregnancy rates (Carrillo-Gonzalez & Maldonado-Estrada, 2020; Xu *et al*, 2020), direct evidence supporting this specific link remains unavailable. Therefore, we have mentioned this possibility only briefly in the discussion.

Methods for Rebuttal Figure 1 and Figure 3

U-¹³C₆-D-Glucose/¹³C₅, ¹⁵N₂-glutamine tracing media preparation for LC-MS

U-¹³C₆-D-Glucose (#CLM-1396-5, Cambridge Isotope Laboratories) or ¹³C₅,¹⁵N₂-glutamine (#CNLM-1275-H-0.1, Cambridge Isotope Laboratories) was supplemented into glucose-free N2B27 basal media -a 1:1 ratio of glucose-free DMEM/F12 (Innovative Research, Inc.) and glucose-free Neurobasal-A (Gibco) supplemented with 1% P/S/G (Gibco), 0.5% N2 (Gibco) and 1% B27 (Gibco)- or into glutamine-free N2B27 basal media -a 1:1 ratio of glutamine-free Advanced DMEM/F12 (Gibco) and glutamine-free Neurobasal (Gibco) supplemented with 1% P/S (Gibco), 0.5% N2 (Gibco) and 1% B27 (Gibco)- to final concentrations of 15 mM and 4 mM, respectively. For J1 cells, 1,000 U/ml mouse leukemia inhibitory factor (mLIF) (Millipore, Merck), 1 μM PD0325901 (Peprtech) and 3 μM CHIR99021 were added to make Naïve tracing media. For PJ1 cells, 1% KnockOut Serum Replacement (Gibco), 12 ng/ml murine bFgf (Peprtech) and 20 ng/ml murine Activin A (Peprtech) were added to create Primed tracing media.

Metabolomics sample preparation for LC-MS

For the intracellular U-¹³C₆-D-Glucose/¹³C₅, ¹⁵N₂-glutamine tracing, 5x10⁵ J1 and PJ1 cells were seeded on 6 well plates. J1 cells cultured in Original Naïve mESC culture media and N2B27 Naïve mESC culture media were seeded on 0.5% porcine gelatin coated 6-well plates in corresponding media. PJ1 cells cultured in Original Primed mESC culture media and N2B27 EpiSC culture media were seeded on Matrigel coated 6-well plates in corresponding media. On the next day of cell seeding, the cells were quickly washed with PBS and then incubated with a medium containing either U-¹³C₆-D-Glucose (15 mM) or ¹³C₅, ¹⁵N₂-glutamine (4 mM). After 24 hours, the medium was removed, and the cells were rapidly washed with ice-cold PBS. Cellular metabolites were extracted using 0.5 mL of 80% methanol (stored at -80 °C for 30 minutes). The cells were scraped, and the metabolite extract was transferred to an Eppendorf tube and cleared by centrifugation (16,000 g for 20 minutes at 4 °C). 200 μL of the extract was taken, the solvent was evaporated using a Speedvac (Genevac, EZ2elite), and the residue was reconstituted with 40 μL of H₂O, followed by 40 μL of methanol. The reconstituted extract was centrifuged again at 16,000 g for

20 minutes at 4 °C, and the supernatant was transferred to LC vials for LC-MS analysis of hydrophilic metabolites and fatty acids separately.

LC-MS analysis

For the analysis of the hydrophilic metabolites, a previously established LC-MS condition was applied (Kang *et al*, 2021). The mobile phases for chromatography were composed of 10 mM ammonium carbonate and 0.05% ammonium hydroxide in water (mobile phase A) and 100% acetonitrile (ACN) (mobile phase B). Five microliters of sample were analyzed by SeQuant® ZIC®-pHILIC LC column (100 x 2.1 mm, 5 µm, 200 Å) coupled with a SeQuant® ZIC®-pHILIC guard column (20 x 2.1 mm). The column temperature was maintained at 30°C, and the elution was performed at a flow rate of 0.25 mL/min with the following gradient: 0-13 min, 80% to 20% B; 13-15 min, 20% B (isocratic); 15-15.1 min, 20% to 80% B; followed by a 5-minute post-run at 80% B. For the fatty acid analysis, 5 µL of each sample was injected into an Acquity UPLC® BEH C18 column (50 x 2.1 mm, 1.7 µm) coupled with an Acquity UPLC® BEH C18 1.7 µm VanGuard Pre-column (5 x 2.1 mm). The column temperature was set to 30°C, and elution occurred at a flow rate of 0.3 mL/min with the following gradient: 0-0.2 min, 5% B; 0.2-1 min, 5-45% B; 1-3 min, 45% to 65% B; 3-3.2 min, 65% to 95% B; 3.2-5 min, 95% B (isocratic), followed by a 3-minute post-run at 5% B. Mass spectrometry analysis was conducted using a Q Exactive Plus (Thermo Fisher Scientific, USA) equipped with a HESI source. MS1 scans were performed in polarity switching mode for non-targeted metabolomics or negative mode for the fatty acids analysis, respectively, with the following settings: sheath gas at 50 arbitrary units, auxiliary gas at 25 arbitrary units, sweep gas at 1 arbitrary unit, spray voltage at 3 kV, capillary temperature at 320°C, and auxiliary gas heater temperature at 350°C. The mass range was set to 70-1050 m/z for hydrophilic conditions and 120-350 m/z for fatty acids, with a resolution of 70,000 and an AGC target of 1×10^6 . Data analysis for metabolomics was carried out using EI Maven v 0.12.0, and metabolite identification was based on retention time and m/z values of in-house library.

References for Rebuttal

- Abu-Elheiga L, Matzuk MM, Kordari P, Oh W, Shaikenov T, Gu Z, Wakil SJ (2005) Mutant mice lacking acetyl-CoA carboxylase 1 are embryonically lethal. *Proc Natl Acad Sci U S A* 102: 12011-12016
- Baker CL, Pera MF (2018) Capturing Totipotent Stem Cells. *Cell Stem Cell* 22: 25-34
- Carrillo-Gonzalez DF, Maldonado-Estrada JG (2020) L-carnitine supplementation in culture media improves the pregnancy rate of in vitro produced embryos with sexed semen from *Bos taurus indicus* cows. *Trop Anim Health Prod* 52: 2559-2565
- Chirala SS, Chang H, Matzuk M, Abu-Elheiga L, Mao J, Mahon K, Finegold M, Wakil SJ (2003) Fatty acid synthesis is essential in embryonic development: fatty acid synthase null mutants and most of the heterozygotes die in utero. *Proc Natl Acad Sci U S A* 100: 6358-6363
- Choi HW, Joo JY, Hong YJ, Kim JS, Song H, Lee JW, Wu G, Scholer HR, Do JT (2016) Distinct Enhancer Activity of Oct4 in Naive and Primed Mouse Pluripotency. *Stem Cell Reports* 7: 911-926
- Cornacchia D, Zhang C, Zimmer B, Chung SY, Fan Y, Soliman MA, Tchieu J, Chambers SM, Shah H, Paull D *et al* (2019) Lipid Deprivation Induces a Stable, Naive-to-Primed Intermediate State of Pluripotency in Human PSCs. *Cell Stem Cell* 25: 120-136 e110
- Gatchalian J, Malik S, Ho J, Lee DS, Kelso TWR, Shokhirev MN, Dixon JR, Hargreaves DC (2018) A non-canonical BRD9-containing BAF chromatin remodeling complex regulates naive pluripotency in mouse embryonic stem cells. *Nat Commun* 9: 5139
- Gentry MS, Roma-Mateo C, Sanz P (2013) Laforin, a protein with many faces: glucan phosphatase, adapter protein, et alii. *FEBS J* 280: 525-537
- Hussein AM, Wang Y, Mathieu J, Margaretha L, Song C, Jones DC, Cavanaugh C, Miklas JW, Mahen E, Showalter MR *et al* (2020) Metabolic Control over mTOR-Dependent Diapause-like State. *Dev Cell* 52: 236-250 e237
- Jiang P, Ren L, Zhi L, Yu Z, Lv F, Xu F, Peng W, Bai X, Cheng K, Quan L *et al* (2021) Negative regulation of AMPK signaling by high glucose via E3 ubiquitin ligase MG53. *Mol Cell* 81: 629-637 e625
- Kang YP, Mockabee-Macias A, Jiang C, Falzone A, Prieto-Farigua N, Stone E, Harris IS, DeNicola GM (2021) Non-canonical Glutamate-Cysteine Ligase Activity Protects against Ferroptosis. *Cell Metab* 33: 174-189 e177
- Kim K-T, Oh J-Y, Park S, Kim S-M, Benjamin P, Park I-H, Chun K-H, Chang Y-T, Cha H-J (2022a) Live isolation of naïve ESCs via distinct glucose metabolism and stored glycogen. *Metabolic Engineering* 72: 97-106
- Kim KT, Oh JY, Park S, Kim SM, Benjamin P, Park IH, Chun KH, Chang YT, Cha HJ (2022b) Live isolation of naive ESCs via distinct glucose metabolism and stored glycogen. *Metab Eng* 72: 97-106
- Kim S-M, Kwon E-J, Kim Y-J, Go Y-H, Oh J-Y, Park S, Do JT, Kim K-T, Cha H-J (2022c) Dichotomous role of Shp2 for naïve and primed pluripotency maintenance in embryonic stem cells. *Stem Cell Research & Therapy* 13: 1-14

- Kinoshita M, Barber M, Mansfield W, Cui Y, Spindlow D, Stirparo GG, Dietmann S, Nichols J, Smith A (2021) Capture of Mouse and Human Stem Cells with Features of Formative Pluripotency. *Cell Stem Cell* 28: 453-471 e458
- Lynch CJ, Bernad R, Martinez-Val A, Shahbazi MN, Nobrega-Pereira S, Calvo I, Blanco-Aparicio C, Tarantino C, Garreta E, Richart-Gines L *et al* (2020) Global hyperactivation of enhancers stabilizes human and mouse naive pluripotency through inhibition of CDK8/19 Mediator kinases. *Nat Cell Biol* 22: 1223-1238
- Martinez-Val A, Lynch CJ, Calvo I, Ximenez-Embun P, Garcia F, Zarzuela E, Serrano M, Munoz J (2021) Dissection of two routes to naive pluripotency using different kinase inhibitors. *Nat Commun* 12: 1863
- Neagu A, van Genderen E, Escudero I, Verwegen L, Kurek D, Lehmann J, Stel J, Dirks RAM, van Mierlo G, Maas A *et al* (2020) In vitro capture and characterization of embryonic rosette-stage pluripotency between naive and primed states. *Nat Cell Biol* 22: 534-545
- Ortmann D, Brown S, Czechanski A, Aydin S, Muraro D, Huang Y, Tomaz RA, Osnato A, Canu G, Wesley BT *et al* (2020) Naive Pluripotent Stem Cells Exhibit Phenotypic Variability that Is Driven by Genetic Variation. *Cell Stem Cell* 27: 470-481 e476
- Pineda CT, Ramanathan S, Fon Tacer K, Weon JL, Potts MB, Ou YH, White MA, Potts PR (2015) Degradation of AMPK by a cancer-specific ubiquitin ligase. *Cell* 160: 715-728
- Qi J, Gong J, Zhao T, Zhao J, Lam P, Ye J, Li JZ, Wu J, Zhou HM, Li P (2008) Downregulation of AMP-activated protein kinase by Cidea-mediated ubiquitination and degradation in brown adipose tissue. *EMBO J* 27: 1537-1548
- Semrau MS, Giachin G, Covaceuszach S, Cassetta A, Demitri N, Storici P, Lolli G (2022) Molecular architecture of the glycogen- committed PP1/PTG holoenzyme. *Nat Commun* 13: 6199
- Stead E, White J, Faast R, Conn S, Goldstone S, Rathjen J, Dhingra U, Rathjen P, Walker D, Dalton S (2002) Pluripotent cell division cycles are driven by ectopic Cdk2, cyclin A/E and E2F activities. *Oncogene* 21: 8320-8333
- Sunaga H, Matsui H, Anjo S, Syamsunarno MR, Koitabashi N, Iso T, Matsuzaka T, Shimano H, Yokoyama T, Kurabayashi M (2016) Elongation of Long-Chain Fatty Acid Family Member 6 (Elovl6)-Driven Fatty Acid Metabolism Regulates Vascular Smooth Muscle Cell Phenotype Through AMP-Activated Protein Kinase/Kruppel-Like Factor 4 (AMPK/KLF4) Signaling. *J Am Heart Assoc* 5
- Suzuki T, Bridges D, Nakada D, Skinnotis G, Morrison SJ, Lin JD, Saltiel AR, Inoki K (2013) Inhibition of AMPK catabolic action by GSK3. *Mol Cell* 50: 407-419
- Wang L, Zhang T, Wang L, Cai Y, Zhong X, He X, Hu L, Tian S, Wu M, Hui L *et al* (2017) Fatty acid synthesis is critical for stem cell pluripotency via promoting mitochondrial fission. *EMBO J* 36: 1330-1347
- White J, Dalton S (2005) Cell cycle control of embryonic stem cells. *Stem Cell Rev* 1: 131-138
- Wu J, Liu J, Thompson I, Oliver CJ, Shenolikar S, Brautigan DL (1998) A conserved domain for glycogen binding in protein phosphatase-1 targeting subunits. *FEBS Lett* 439: 185-191
- Xu H, Jia C, Cheng W, Zhang T, Tao R, Ma Y, Si L, Xu Y, Li J (2020) The Effect of L-Carnitine Additive During In Vitro Maturation on the Vitrification of Pig Oocytes. *Cell Reprogram* 22: 198-207

Yan H, Malik N, Kim YI, He Y, Li M, Dubois W, Liu H, Peat TJ, Nguyen JT, Tseng YC *et al* (2021) Fatty acid oxidation is required for embryonic stem cell survival during metabolic stress. *EMBO Rep* 22: e52122

Zhu Y, Zhang M, Kelly AR, Cheng A (2014) The carbohydrate-binding domain of overexpressed STBD1 is important for its stability and protein-protein interactions. *Biosci Rep* 34

Dear Keuntae,

Thank you for submitting your revised manuscript. It has now been seen by two of the original referees. I apologize for this unusual delay in getting back to you, it took longer than anticipated to receive the referee reports.

As you can see, the referees find that the study is significantly improved during revision and recommend publication. However, I need you to address the points below before I can accept the manuscript.

- Please address the remaining concerns of referee #2.
- In the Data Availability section, please provide a URL which directly resolves to GSE272593.
- Please rename the Competing interest section as Disclosure Statement and Competing Interests.
- We note a mismatch between the manuscript text and our manuscript tracking system in terms of funding information - grant numbers - RS-2024-00432867 is provided in the manuscript while RS-2022-00070316 is entered in the manuscript tracking system.
- We note that Figures 2, 5 and 6 span across two pages each, which is not allowed as per journal format requirement. Please rearrange the figures as one page per figure.
- We note that the 'Supplemental Table' is a dataset. Thus, it should be resubmitted as such, and source file name, title and manuscript callouts need to be updated as "Dataset EV1".
- We note that Fig. 6J and the suppl. Table (Dataset EV1 to be as per the above point) are currently not called out in the text.
- As for the Supplementary Information file, its nomenclature needs to be updated to Appendix Figure S1-S6 including all instances of the word "Supplemental"; a brief ToC should be provided on the title page that shows page numbers of each figure; each figure legend should go after each Appendix figure.
- All research articles submitted as revised versions must include a structured methods section that includes a Reagents and Tools Table followed by a Methods and Protocols section. Please see <https://www.embopress.org/page/journal/14693178/authorguide#structuredmethods> for further information.
- We note that some of the source data is submitted in the format of .pzfx, which is not in the list of allowed formats. Please see the information sent by our Source Data Coordinator Dr. Hannah Sonntag ("Numerical data can be provided as individual .xls (including a tab describing the data) or as .csv files (including a separate README file). Small scale imaging data should be provided in a common file format that does not change the resolution or alter the image. We recommend using .tiff format.")
- We note that phosphoproteome data from Martinez-Val et al., 2021 was re-analyzed in this study and the results were presented in Figure 1B-E. I presume that the referred dataset is PXD018694, please confirm. Moreover, please cite this dataset in the form of data citation, too. I paste an example below, please see <https://www.embopress.org/page/journal/14693178/authorguide#referencesformat> for further information.

In-text citation: "...were grouped based on the relative levels of AR-Vs expressed, mainly AR-V7 (Hörnberg et al, 2011; Data ref: Hörnberg et al, 2011)."

In the reference list:

- Hörnberg E, Ylitalo EB, Crnalic S, Antti H, Stattin P, Widmark A, Bergh A, Wikström P (2011) Gene Expression Omnibus GSE29650 (<https://www.ncbi.nlm.nih.gov/geo/query/acc.cgi?acc=GSE29650>). [DATASET]
- Hörnberg E, Ylitalo EB, Crnalic S, Antti H, Stattin P, Widmark A, Bergh A, Wikström P (2011) Expression of androgen receptor splice variants in prostate cancer bone metastases is associated with castration-resistance and short survival. PLoS One 6: e19059
- Materials and Methods should be renamed as Methods.
 - The order of the sections should be as follows: Title page - Abstract & Keywords - Introduction - Results - Discussion - Methods - Data Availability - Acknowledgments - Disclosure Statement & Competing Interests - References - Figure Legends - (Main Tables with legends if applicable) - Expanded View Figure Legends.
 - We note a potential image reuse between Figure 2i and Figure 3C and a blot reuse between Figure 3B and Appendix Figure S4G, which are only allowed if the reused panels are derived from the same experiment, in which case this needs to be clarified in all respective figure legends.
 - Our production/data editors have asked you to clarify several points in the figure legends:
 - o Please note that the exact p values are not provided in the legends of figures 3G, 4D, G; 5B, F, I; 6D, L; supplementary figure(s) 4B, 6J.
 - o Please indicate the statistical test used for data analysis in the legends of figures 1C, 5I, 6D; Supplementary figure(s) 1A; 4A.
 - o Please note that in figures 5I there is a mismatch between the annotated p values in the figure legend and the annotated p values in the figure file that should be corrected.
 - o Please note that for the figures 2E, H, J; 4C, F; 6E, G, L, M; Supplementary figure(s) 4D, F ""*****/**/*/*"" has not been represented in the figures. If it represents P value, please indicate the statistical test and exact p values used for data analysis. Rectify this in the figures or legends as applicable.
 - o Please note that information related to n is missing in the legends of figures 1C, 2E, H, J; 4C, F; 6E, G, M; Supplementary figure(s) 1A, 2A, B; 4A, C, D, F.
 - o Although 'n' is provided, please describe the nature of entity for 'n' in the legends of figures 3G, 4D, G; 5B, F, I; 6D, L;

Supplementary figures(s) 4B, 6J.

o Please note that the error bars are not defined in the legends of figures 2E, H, J; 3G, 4C, D, F, G; 5B, F, I; 6D, E, G, L, M; Supplementary figure(s) 2B, 4A, C, D, F; 6J."

o Please note that scale bar and its definition are missing for supplementary figure 2A.

o Please note that the asterisk is not defined in the legend of supplementary figure 6G. This needs to be rectified.

o Please note that the white arrows are not defined in the legend of figure 2I. This needs to be rectified.

• Papers published in EMBO Reports include a 'synopsis' and 'bullet points' to further enhance discoverability. Both are displayed on the html version of the paper and are freely accessible to all readers. The synopsis includes a short standfirst summarizing the study in 1 or 2 sentences (max 35 words) that summarize the paper and are provided by the authors and streamlined by the handling editor. I would therefore ask you to include your synopsis blurb and 3-5 bullet points listing the key experimental findings.

Thank you again for giving us to consider your manuscript for EMBO Reports, I look forward to your minor revision.

Kind regards,

Deniz

--

Deniz Senyilmaz Tiebe, PhD
Senior Scientific Editor
EMBO Reports

Referee #1:

The revised manuscript is significantly more solid than the submitted version.

Several technical issues raised by reviewers have been addressed. Also the statistical analyses are more sound.

I find the revised manuscript suitable for publication.

Referee #2:

In this revised manuscript, the authors made extensive revision and the quality of the manuscript is significantly improved. Now I agree with the publication of the present version after the revision for the points in below.

Line 35: The pluripotent stem cells in the blastocysts should not be designated as ESCs. 'Embryonic stem cells' is the name of the cell line. It should be designated as (naïve) pluripotent stem cells or epiblasts.

Line 46: Is MEK1 a sole target of PD032591? PD032591 is a dual inhibitor of MEK1 and MEK2, and both are functional in ESCs (Choi et al, Nature, 2017).

Kim & Kwon et al. EMBOR-2024-59124V3_Response to editor and reviewer comments**[Response to the editor]**

We deeply appreciate the reviewers and editor's careful comments on our revised manuscript. Below, we responded point-by-point to each comment and uploaded the most updated version of files through the system. Once again, we truly thank to all the precious feedback which significantly improved our manuscript for publication in EMBO Reports journal.

[Editor comments]

- Please address the remaining concerns of referee #2.

: As per the reviewer commented, we revised the main text according to each point.

- In the Data Availability section, please provide a URL which directly resolves to GSE272593.

: As per the editor suggested, we provided proper URL to access our dataset in the revised manuscript.

- Please rename the Competing interest section as Disclosure Statement and Competing Interests.

: We changed this accordingly.

- We note a mismatch between the manuscript text and our manuscript tracking system in terms of funding information - grant numbers - RS-2024-00432867 is provided in the manuscript while RS-2022-00070316 is entered in the manuscript tracking system.

: We checked the grant number again and found that RS-2024-00432867 is the correct number. We revised this information at tracking system accordingly.

Kim & Kwon et al. EMBOR-2024-59124V3_Response to editor and reviewer comments

- We note that Figures 2, 5 and 6 span across two pages each, which is not allowed as per journal format requirement. Please rearrange the figures as one page per figure.

: We resized figures 2, 5 and 6 to make a single page in the revised version.

- We note that the 'Supplemental Table' is a dataset. Thus, it should be resubmitted as such, and source file name, title and manuscript callouts need to be updated as "Dataset EV1".

: We changed the "Supplement Table" into "Dataset EV". We also changed main text accordingly.

- We note that Fig. 6J and the suppl. Table (Dataset EV1 to be as per the above point) are currently not called out in the text.

: Thanks for pointing out this. We cited in text accordingly.

- As for the Supplementary Information file, its nomenclature needs to be updated to Appendix Figure S1-S6 including all instances of the word "Supplemental"; a brief ToC should be provided on the title page that shows page numbers of each figure; each figure legend should go after each Appendix figure.

: We updated the nomenclature for Figure EV# (Supplementary Figures in previous version accordingly and revised the separate document for each figure legend.

- All research articles submitted as revised versions must include a structured methods section that includes a Reagents and Tools Table followed by a Methods and Protocols section. Please see <https://www.embopress.org/page/journal/14693178/authorguide#structuredmethods> for further information.

: We completed the Reagents and Tools Table accordingl to the instruction by uploading as a separate file.

Kim & Kwon et al. EMBOR-2024-59124V3_Response to editor and reviewer comments

- We note that some of the source data is submitted in the format of .pzfx, which is not in the list of allowed formats. Please see the information sent by our Source Data Coordinator Dr. Hannah Sonntag ("Numerical data can be provided as individual .xls (including a tab describing the data) or as .csv files (including a separate README file). Small scale imaging data should be provided in a common file format that does not change the resolution or alter the image. We recommend using .tiff format.")

: We have reformatted the source data previously provided in .pzfx format into .xlsx files and converted all figure formats to .tiff and uploaded to the system.

- We note that phosphoproteome data from Martinez-Val et al., 2021 was re-analyzed in this study and the results were presented in Figure 1B-E. I presume that the referred dataset is PXD018694, please confirm. Moreover, please cite this dataset in the form of data citation, too. I paste an example below, please see <https://www.embopress.org/page/journal/14693178/authorguide#referencesformat> for further information.

In-text citation: "...were grouped based on the relative levels of AR-Vs expressed, mainly AR-V7 (Hörnberg et al, 2011; Data ref: Hörnberg et al, 2011)."

In the reference list:

Hörnberg E, Ylitalo EB, Crnalic S, Antti H, Stattin P, Widmark A, Bergh A, Wikström P (2011) Gene Expression Omnibus GSE29650 (<https://www.ncbi.nlm.nih.gov/geo/query/acc.cgi?acc=GSE29650>). [DATASET]

Hörnberg E, Ylitalo EB, Crnalic S, Antti H, Stattin P, Widmark A, Bergh A, Wikström P (2011) Expression of androgen receptor splice variants in prostate cancer bone metastases is associated with castration-resistance and short survival. PLoS One 6: e19059

: We cited the Martinez-Val et al study accordingly based on the format suggested above (sentence 98-99 and sentence 629-630 in the Main text).

Kim & Kwon et al. EMBOR-2024-59124V3_Response to editor and reviewer comments

- Materials and Methods should be renamed as Methods.

: We changed the section name properly.

- The order of the sections should be as follows: Title page - Abstract & Keywords - Introduction - Results - Discussion - Methods - Data Availability - Acknowledgments - Disclosure Statement & Competing Interests - References - Figure Legends - (Main Tables with legends if applicable) - Expanded View Figure Legends.

: We changed the order of section accordingly.

- We note a potential image reuse between Figure 2i and Figure 3C and a blot reuse between Figure 3B and Appendix Figure S4G, which are only allowed if the reused panels are derived from the same experiment, in which case this needs to be clarified in all respective figure legends.

: Thanks for pointing out this. Those figures are derived from same batch of experiments, which we indicated at all respective figure legend in this updated version.

- Our production/data editors have asked you to clarify several points in the figure legends:

: We appreciate the editor's points regarding the missing information in figure legend section. Below, we provide the corresponding point-by-point response along with the necessary rectifications.

o Please note that the exact p values are not provided in the legends of figures 3G, 4D, G; 5B, F, I; 6D, L; supplementary figure(s) 4B, 6J.

: We added exact p values for all above figures in corresponding figure legend or each figure panel, respectively.

Kim & Kwon et al. EMBOR-2024-59124V3_Response to editor and reviewer comments

o Please indicate the statistical test used for data analysis in the legends of figures 1C, 5I, 6D; Supplementary figure(s) 1A; 4A.

: We added detail statistical test used for each dataset in this updated version.

o Please note that in figures 5I there is a mismatch between the annotated p values in the figure legend and the annotated p values in the figure file that should be corrected.

: We revised the p value information in figure legend.

o Please note that for the figures 2E, H, J; 4C, F; 6E, G, L, M; Supplementary figure(s) 4D, F ""****/**/**/*"" has not been represented in the figures. If it represents P value, please indicate the statistical test and exact p values used for data analysis. Rectify this in the figures or legends as applicable.

: We revised the p value information in both figure legend and corresponding figure panels.

o Please note that information related to n is missing in the legends of figures 1C, 2E, H, J; 4C, F; 6E, G, M; Supplementary figure(s) 1A, 2A, B; 4A, C, D, F.

: We revised statistical replication information in all figures accordingly.

o Although 'n' is provided, please describe the nature of entity for 'n' in the legends of figures 3G, 4D, G; 5B, F, I; 6D, L; Supplementary figures(s) 4B, 6J.

: We rectified all the statistical replication information, accordingly.

o Please note that the error bars are not defined in the legends of figures 2E, H, J; 3G, 4C, D, F, G; 5B, F, I; 6D, E, G, L, M; Supplementary figure(s) 2B, 4A, C, D, F; 6J."

: We added the definition of error bar for all the figures in corresponding figure legend section.

o Please note that scale bar and its definition are missing for supplementary figure 2A.

: We added the proper scale bars in figure panels and information at figure legend section.

o Please note that the asterisk is not defined in the legend of supplementary figure 6G. This needs to be rectified.

Kim & Kwon et al. EMBOR-2024-59124V3_Response to editor and reviewer comments

: We added the definition of asterisk in corresponding legendary section.

o Please note that the white arrows are not defined in the legend of figure 2I. This needs to be rectified.

: We added the definition of white arrows at corresponding figure legend.

- Papers published in EMBO Reports include a 'synopsis' and 'bullet points' to further enhance discoverability. Both are displayed on the html version of the paper and are freely accessible to all readers. The synopsis includes a short standfirst summarizing the study in 1 or 2 sentences (max 35 words) that summarize the paper and are provided by the authors and streamlined by the handling editor. I would therefore ask you to include your synopsis blurb and 3-5 bullet points listing the key experimental findings.

: We provided the synopsis and bullet points for our study by uploading through the system as a separate file.

[Reviewer comments]

Referee #1:

The revised manuscript is significantly more solid than the submitted version.

Several technical issues raised by reviewers have been addressed. Also the statistical analyses are more sound.

I find the revised manuscript suitable for publication.

: We appreciate the reviewer's positive feedback.

Referee #2:

In this revised manuscript, the authors made extensive revision and the quality of the manuscript is significantly improved. Now I agree with the publication of the present version after the revision for the points in below.

: We appreciate the reviewer's positive feedback and comments. Below, we made our response accordingly.

Line 35: The pluripotent stem cells in the blastocysts should not be designated as ESCs. 'Embryonic stem cells' is the name of the cell line. It should be designated as (naïve) pluripotent stem cells or epiblasts.

: Thanks for the reviewer's critical point. As per the reviewer commented, we revised naïve ESC into naïve pluripotent stem cells.

Line 46: Is MEK1 a sole target of PD032591? PD032591 is a dual inhibitor of MEK1 and MEK2, and both are functional in ESCs (Choi et al, Nature, 2017).

: As per the reviewer pointed, we changed the MEK1 into MEK1/2 in order to indicated the target of PD0325901 is both MEK1 and MEK2.

Dr. Keun-Tae Kim
Seoul National University, College of Pharmacy
Department of Pharmacy
1 Gwanak-ro Gwanak-gu
Seoul, Seoul 08826
Korea, Republic of

Dear Keuntae,

Thank you for submitting your revised manuscript. I have now looked at everything and all is fine. Therefore, I am very pleased to accept your manuscript for publication in EMBO Reports.

Congratulations on a nice work!

Kind regards,

Deniz Senyilmaz Tiebe

--

Deniz Senyilmaz Tiebe, PhD
Senior Scientific Editor
EMBO Reports

--
